# Global and context-specific transcriptional consequences of oncogenic Fbw7 mutations

H Nayanga Thirimanne[1,2,3], Feinan Wu[4], Derek H Janssens[5], Jherek Swanger[1,2], Ahmed Diab[1,2], Heather M Feldman[2], Robert A Amezquita[6], Raphael Gottardo[6], Patrick J Paddison[2], Steven Henikoff[5,7,8]*, Bruce E Clurman[1,2,3,9]*

[1]Clinical Research Division, Fred Hutchinson Cancer Research Center, Seattle, United States; [2]Human Biology Division, Fred Hutchinson Cancer Research Center, Seattle, United States; [3]Department of Pathology, University of Washington, Seattle, United States; [4]Genomics and Bioinformatics Resource, Fred Hutchinson Cancer Research Center, Seattle, United States; [5]Basic Science Division, Fred Hutchinson Cancer Research Center, Seattle, United States; [6]Vaccine and Infectious Disease Division, Fred Hutchinson Cancer Research Center, Seattle, United States; [7]Genome Sciences, University of Washington, Seattle, United States; [8]Howard Hughes Medical Institute, Chevy Chase, United States; [9]Department of Medicine, University of Washington, Seattle, United States

*For correspondence:
steveh@fhcrc.org (SH);
bclurman@fredhutch.org (BEC)

Competing interest: The authors declare that no competing interests exist.

**Abstract** The Fbw7 ubiquitin ligase targets many proteins for proteasomal degradation, which include oncogenic transcription factors (TFs) (e.g., c-Myc, c-Jun, and Notch). Fbw7 is a tumor suppressor and tumors often contain mutations in *FBXW7*, the gene that encodes Fbw7. The complexity of its substrate network has obscured the mechanisms of Fbw7-associated tumorigenesis, yet this understanding is needed for developing therapies. We used an integrated approach employing RNA-Seq and high-resolution mapping (cleavage under target and release using nuclease) of histone modifications and TF occupancy (c-Jun and c-Myc) to examine the combinatorial effects of misregulated Fbw7 substrates in colorectal cancer (CRC) cells with engineered tumor-associated *FBXW7* null or missense mutations. Both Fbw7 mutations caused widespread transcriptional changes associated with active chromatin and altered TF occupancy: some were common to both Fbw7 mutant cell lines, whereas others were mutation specific. We identified loci where both Jun and Myc were coregulated by Fbw7, suggesting that substrates may have synergistic effects. One coregulated gene was *CIITA*, the master regulator of MHC Class II gene expression. Fbw7 loss increased MHC Class II expression and Fbw7 mutations were correlated with increased CIITA expression in TCGA colorectal tumors and cell lines, which may have immunotherapeutic implications for Fbw7-associated cancers. Analogous studies in neural stem cells in which *FBXW7* had been acutely deleted closely mirrored the results in CRC cells. Gene set enrichment analyses revealed Fbw7-associated pathways that were conserved across both cell types that may reflect fundamental Fbw7 functions. These analyses provide a framework for understanding normal and neoplastic context-specific Fbw7 functions.

## Editor's evaluation

Fbw7 functions to control the abundance of more than 2 dozen transcriptional regulators, but how this affects transcription at the global level is largely unknown. The authors employ RNA-Seq, CUT&RUN on H3K27ac/H3K27me3, and a detailed analysis of the loci affected to provide a global

analysis of the effect of Fbw7 mutation on transcription in HCT116 cells as well as neural stem cells. The results reveal complex, but intriguing, results suggesting that Fbw7 mutation affects primarily Jun and Myc functions in distal regulatory regions rather than target gene promoters. Although HCT116 cells employed (WT, Fbw7-/-, and Fbw7R/+) are clonal, there is significant overlap in the two mutant lines, which suggests that a substantial fraction of the effects reflect loss of Fbw7 activity. Analogous patterns related to Jun and Myc levels at distal regulatory regions are seen in the neural stem cells, where there is a pool of depleted cells rather than clonal cells derived from targeted mutagenesis. Intriguingly, gene sets related to epithelial mesenchymal transition (EMT) were enriched in the upregulated transcripts in both Fbw7-/- and R/+ mutant cells consistent with the idea that Fbw7 targets EMT regulatory proteins for degradation. Additional experiments and analyses performed during revision substantially strengthened this paper.

## Introduction

SCFs (Skp1-Cul1-F-box proteins) are multi-subunit ubiquitin ligases that target proteins for degradation through the conjugation of polyubiquitin chains that signal their destruction by the proteasome (*Deshaies and Joazeiro, 2009*; *Lee and Diehl, 2014*). F-box proteins are SCF substrate receptors and often target proteins for ubiquitylation in response to substrate modifications (*Skaar et al., 2013*; *Yumimoto and Nakayama, 2020*). The Fbw7 F-box protein binds to substrates after they become phosphorylated within conserved phosphodegron motifs that mediate high-affinity interactions with the Fbw7 β-propeller (*Davis et al., 2014*; *Hao et al., 2007*; *Nash et al., 2001*; *Orlicky et al., 2003*; *Yumimoto and Nakayama, 2020*). Like its yeast orthologues, Fbw7 functions as a dimer, which influences substrate ubiquitylation and substrate binding (*Hao et al., 2007*; *Kominami et al., 1998*; *Tang et al., 2007*; *Welcker et al., 2013*; *Welcker and Clurman, 2007*; *Zhang and Koepp, 2006*). The Fbw7 F-box binds to Skp1, which allows Fbw7 to bring phosphorylated substrates into proximity with the remainder of the SCF complex. Phosphodegrons vary in their affinity for Fbw7, and this influences substrate binding. High-affinity degrons are sufficient to enable productive binding to monomeric Fbw7, whereas lower affinity substrates bind through the concerted interaction of two degrons, each with one protomer of an Fbw7 dimer (*Welcker et al., 2013*; *Welcker et al., 2022*).

Approximately 30 Fbw7 substrates are known. Most are broadly acting transcription factors (TFs) that control processes such as proliferation, differentiation, and metabolism, and these include c-Myc, Notch, c-Jun, PGC-1α, SREBP1/2, and others (*Cremona et al., 2016*; *Davis et al., 2014*; *Welcker and Clurman, 2008*; *Yumimoto and Nakayama, 2020*). Fbw7 also targets other proteins, most notably cyclin E and MCL-1. Fbw7 exerts its cellular functions through the combined regulation of its many substrates, and different cell types express subsets of substrates that are targeted for degradation by Fbw7 only after they acquire specific phosphorylation. Some substrates (cyclin E and SREBP) are highly Fbw7 dependent, whereas others (e.g., Myc, Jun, and Notch) have multiple turnover pathways and are only regulated by Fbw7 in specific contexts (*Carrieri and Dale, 2016*; *Lopez-Bergami et al., 2010*). In most cases, glycogen synthase kinase 3β (GSK-3β) is one of the degron kinases, which may allow Fbw7 to coordinately couple its substrate network to mitogenic signals. Some TFs are phosphorylated when they are bound to their target genes (*Fryer et al., 2004*; *Punga et al., 2006*), providing another control over their susceptibility to Fbw7-mediated degradation. This complexity has made it difficult to fully ascertain Fbw7 functions, which is compounded by the fact that most substrates are TFs that regulate gene networks themselves.

Several Fbw7 substrates are oncoproteins, such as c-Myc, cyclin E, c-Jun, and Notch1. Fbw7 loss deregulates these oncoproteins and *FBXW7* is a commonly mutated tumor suppressor gene (*Davis et al., 2014*; *Shimizu et al., 2018*; *Tan et al., 2008*; *Yeh et al., 2018*; *Yumimoto and Nakayama, 2020*). The most frequent mutations are heterozygous missense mutations, hereafter termed Fbw7$^{R/+}$, that target one of the three arginine residues that form Fbw7's substrate-binding pocket. Fbw7$^{R/+}$ weaken substrate binding and are thought to act as dominant negatives by forming impaired Fbw7$^R$/Fbw7$^{WT}$ heterodimers (*Hao et al., 2007*; *Welcker et al., 2013*; *Welcker and Clurman, 2007*). While Fbw7$^{R/+}$ mutations are common, Fbw7$^{+/-}$ mutations are not, suggesting that Fbw7$^{R/+}$ mutations are not simple loss-of-function mutations, and this is supported by Fbw7$^{R/+}$ mouse models, which develop tumors to a greater extent than Fbw7$^{+/-}$ mice (*Davis et al., 2014*). Fbw7$^{R/+}$ may preferentially disrupt the aspects of Fbw7 function that depend upon fully functional Fbw7 dimers. The 'just enough' model

posits that Fbw7$^{R/+}$ reduces activity sufficiently to impair tumor suppressor functions while preserving beneficial Fbw7 activities (*Davis and Tomlinson, 2012*). In addition to Fbw7$^{R/+}$, canonical biallelic loss-of-function Fbw7$^{-/-}$ mutations (e.g., nonsense, truncations, frame shifts, and deletions) also occur in tumors. Different cancers have different mutational spectra; T-cell acute lymphocytic leukemias (T-ALLs) have almost exclusively Fbw7$^{R/+}$ whereas colorectal cancers (CRCs) have both Fbw7$^{R/+}$ and Fbw7$^{-/-}$ mutations. In all cases, tumorigenesis associated with Fbw7 mutations likely involves the concerted activities of multiple oncogenic substrates.

We assessed the global transcriptional consequences of oncogenic Fbw7 mutations by using RNA-Seq and high-resolution mapping of histone modifications and oncogenic TF (c-Jun and c-Myc, here onwards Jun and Myc) occupancy in isogenic Hct116 CRC cells with engineered Fbw7$^{-/-}$ and Fbw7$^{R/+}$ mutations. Both mutations caused widespread, yet highly context-specific transcriptional changes associated with active chromatin and altered TF occupancy. Many deregulated genes and loci were shared between the two mutant cell lines, and the consequences of Fbw7$^{-/-}$ were generally greater than Fbw7$^{R/+}$. While both mutations impacted small subsets of mapped Jun and Myc loci, there was substantial overlap, and Jun and Myc were coregulated by Fbw7 at these shared binding sites. One coregulated gene was *CIITA* (Class II Major Histocompatibility Complex Transactivator), the master regulator of MHC Class II gene expression (*Masternak et al., 2000*; *Reith et al., 2005*). Jun and Myc occupancy upstream of *CIITA* were increased in Fbw7$^{-/-}$ cells, leading to inappropriate MHC Class II RNA and protein expression. Analyses of TCGA CRC and cell lines further correlated Fbw7 mutations with MHC Class II gene expression, which may have important prognostic and therapeutic implications for Fbw7-associated CRCs. Because Fbw7 regulates neural stem cells (NSCs) (*Hoeck et al., 2010*) and Fbw7 expression is repressed in glioblastomas (*Hagedorn et al., 2007*), we studied NSCs in which Fbw7 was acutely deleted as an orthogonal system. The consequences of Fbw7 loss in NSCs closely mirrored the Hct116 results in the extent and patterns of transcriptional deregulation. Gene set enrichment analyses of Fbw7-dependent genes revealed extensive conservation between NSCs and Hct116 with respect to biologic processes, which included epithelial-mesenchymal transition (EMT) and MHC Class II complex. Overall, these data establish a framework for understanding the mechanisms of Fbw7 function and tumor suppression.

## Results

### Fbw7 null and missense mutations lead to distinct gene expression profiles

We previously engineered a panel of isogenic Hct116 cells by mutating the endogenous wild-type (WT) *FBXW7* locus to either a heterozygous Fbw7$^{R505C/+}$ (Fbw7$^{R/+}$) or a homozygous null (Fbw7$^{-/-}$) mutation (*Figure 1A*; *Davis et al., 2018*; *Grim et al., 2008*). We performed RNA sequencing to identify the global transcriptome changes arising in both cell lines. Principal component analysis (PCA) revealed that the Fbw7$^{R/+}$ and Fbw7$^{-/-}$ cells clustered apart from one another, indicating that the two mutations have distinct effects on the transcriptome relative to WT cells (*Figure 1—figure supplement 1*). Compared with WT cells, 11.3% and 5.4% of the captured protein-coding genes were differentially expressed (DE) in Fbw7$^{-/-}$ and Fbw7$^{R/+}$ cells, respectively. Some genes were DE in both Fbw7$^{-/-}$ and Fbw7$^{R/+}$ cells, whereas others were unique to one cell line (*Figure 1B*, *Figure 1—source data 1*). Hierarchical clustering of the DE protein-coding genes identified transcripts that were: (1) upregulated (cluster 1) or downregulated (cluster 2) in just Fbw7$^{-/-}$ cells, (2) genes upregulated (cluster 6) and downregulated (cluster 5) in just Fbw7$^{R/+}$ cells, and (3) genes that show similar expression changes in response to both Fbw7 mutations (clusters 3 and 4) (*Figure 1C*).

We examined the Fbw7-dependent upregulated or downregulated transcripts for enrichment of gene sets representing cellular pathways and processes using the Enrichr enrichment analysis web-based tool (*Figure 1D*; *Chen et al., 2013*; *Kuleshov et al., 2016*). *Figure 1D* highlights some significantly enriched gene sets representing TF targets (TRANSFAC & JASPAR), gene ontologies (GOs), and pathways (MSigDB). *Figure 1—source data 3* contains the full listing of enriched terms and their constituent genes, for both Fbw7$^{-/-}$ and Fbw7$^{R/+}$ cells. Most enriched gene sets were common to both Fbw7$^{-/-}$ and Fbw7$^{R/+}$ cells and some likely reflect known Fbw7 properties, such as the p53-p21 pathway. Fbw7 loss causes p53 induction in cell lines and tumors, and also impacts the cell cycle through substrates like cyclin E (*Li et al., 2015*; *Minella et al., 2007*; *Minella et al., 2002*).

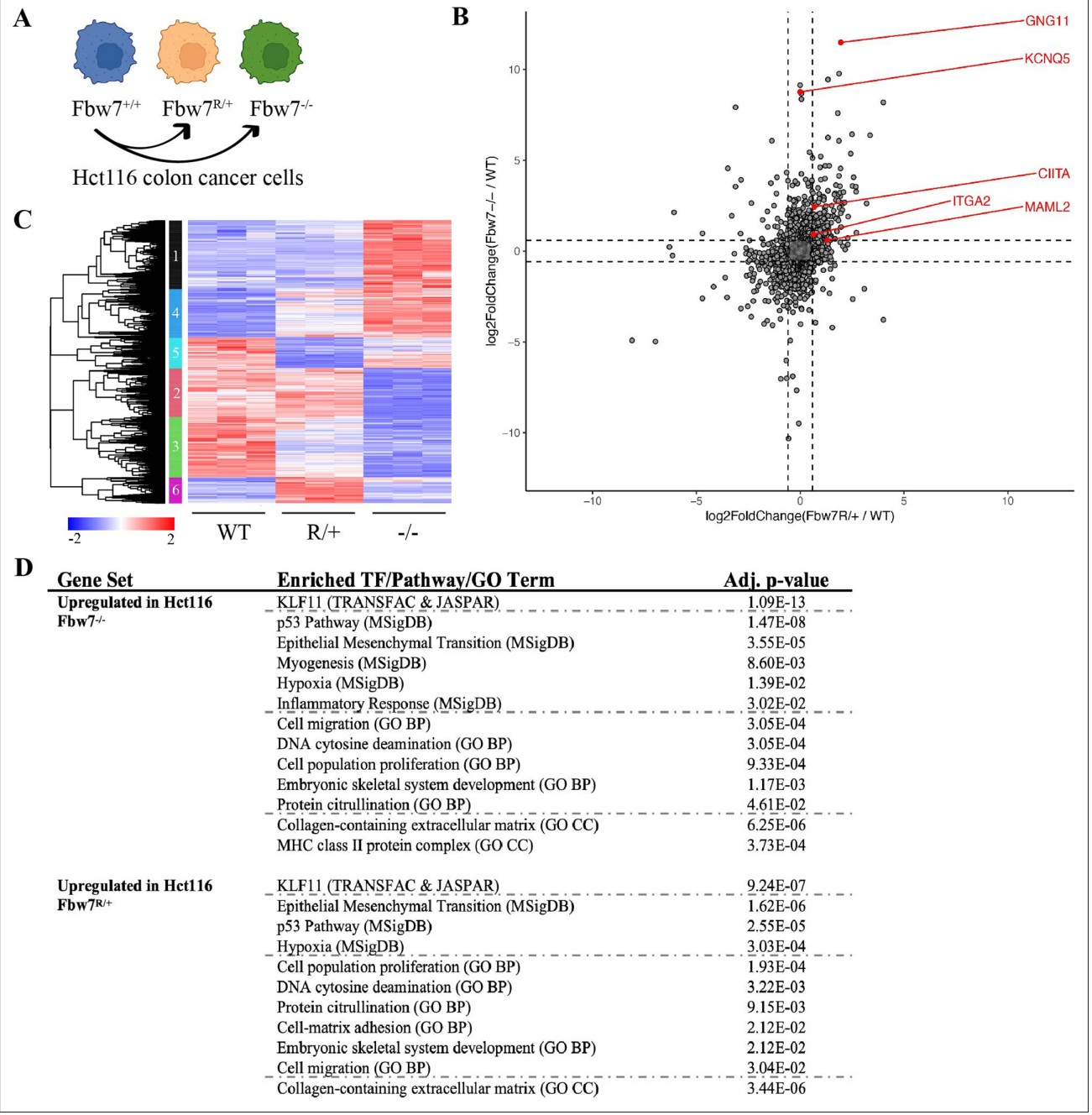

**Figure 1.** RNA-Seq reveals differential gene expression in Hct116 Fbw7$^{-/-}$ and Fbw7$^{R/+}$ cells. (**A**) Genetically engineered isogenic cell lines used in the study: Hct116 wild-type (WT), Fbw7$^{-/-}$, and Fbw7$^{R/+}$. (**B**) Differentially expressed (DE) protein-coding genes (represented by each dot) in Fbw7$^{-/-}$ or Fbw7$^{R/+}$ (FDR<0.05). Dashed lines mark log$_2$FC=0.6. (**C**) Hierarchical clustering of DE protein-coding genes. The heatmap shows the intensity of expression of each gene (y-axis) for three replicates per cell type (x-axis). Three replicates per cell type were included. Replicates for each genotype were from a single clone, however from separately cultured samples. (**D**) Transcription factors, pathways (MSigDB), and GO terms that were enriched in genes upregulated in Fbw7$^{-/-}$ and Fbw7$^{R/+}$ Hct116 cells. Detailed output of differential expression analysis, hierarchical clustering, and Enrichr analysis are provided as *Figure 1—source data 1*, *Figure 1—source data 2*, *Figure 1—source data 3*, respectively. See *Figure 1—figure supplement 1* for the PCA of Hct116 RNA-Seq and *Figure 1—figure supplement 2* for cell proliferation data. GO, gene ontology; PCA, principal component analysis.

The online version of this article includes the following source data and figure supplement(s) for figure 1:

**Source data 1.** Differential expression analysis of Hct116 RNA-Seq.

**Source data 2.** Hierarchical cluster output file.

**Source data 3.** Enrichr output for Hct116 differentially expressed genes.

*Figure 1 continued on next page*

*Figure 1 continued*

**Figure supplement 1.** Principal component analysis (PCA) of RNA-Seq from Hct116 cells.

**Figure supplement 2.** Proliferation of Fbw7 mutant Hct116 cells.

Accordingly, p21 mRNA expression was increased in Fbw7$^{-/-}$ and Fbw7$^{R/+}$ Hct116 cells (2.0-fold and 2.3-fold, respectively, *Figure 1—source data 1*), and both cell lines exhibited a modest increase in doubling time and S-phase fraction (*Figure 1—figure supplement 2*). Gene sets related to EMT were enriched in the upregulated transcripts in Fbw7$^{-/-}$ and Fbw7$^{R/+}$ cells, consistent with findings that Fbw7 targets proteins that control EMT regulators such as Zeb2, Snail, and LSD-1 (*Diaz and de Herreros, 2016*; *Lan et al., 2019*; *Li et al., 2019*; *Zhang et al., 2018*). Other shared enriched genes sets may also be related to EMT, such as extracellular matrix (ECM) and cell migration. Some of these same enriched gene sets were found in downregulated transcripts in Fbw7$^{-/-}$ and Fbw7$^{R/+}$ cells, which also included pathways related to NF-kB signaling and inflammation. MHC Class II genes were enriched in only Fbw7$^{-/-}$ cells and are studied in more detail below.

We also examined the DE genes for putative regulatory TFs that are Fbw7 substrates. Intriguingly, gene sets associated with binding sites for KLF proteins, a TF family that is broadly targeted by Fbw7, were highly enriched in the upregulated differential genes in both cell types, but not in the downregulated genes (*Liu et al., 2010*; *Yu et al., 2018*; *Yumimoto and Nakayama, 2020*; *Zhao et al., 2010*; *Zhao and Sun, 2013*). Although there were gene clusters that were uniquely deregulated in Fbw7$^{R/+}$ cells (clusters 5 and 6, *Figure 1C*), we did not find any Fbw7$^{R/+}$-specific enriched gene sets.

## Chromatin regulation in Fbw7 mutant cells

We next studied how Fbw7 mutations globally influence chromatin marks and whether specific TF substrates were implicated in Fbw7-dependent active chromatin. Histone H3 lysine-27 acetylation (H3K27ac) and Histone H3 lysine-27 trimethylation (H3K27me3) provide simple readouts of transcriptionally active versus repressive chromatin, respectively (*Karlić et al., 2010*). We used cleavage under target and release using nuclease (CUT&RUN) (*Janssens et al., 2018*; *Skene et al., 2018*; *Skene and Henikoff, 2017*) to obtain high-resolution maps of H3K27ac and H3K27me3 in each of the Hct116 cell lines (*Figure 2—figure supplement 1*). As expected, the H3K27ac signal within the 2-kb region flanking the transcriptional start sites (TSSs) of genes was positively correlated with their expression (r=0.44, p<2.2e−16), whereas the amount of H3K27me3 was negatively correlated (r=−0.22, p<2.2e−16) (*Figure 2A*). For example, the *GNG11* gene, whose expression is upregulated in Fbw7$^{-/-}$ cells, contains increased H3K27ac and decreased H3K27me3, compared with WT (*Figures 1B and 2B*).

Genome-wide analysis identified sites with increased H3K27ac in Fbw7 mutant cells (Fbw7$^{-/-}$: 9.4%, Fbw7$^{R/+}$: 7.6%) compared with control cells, as well as sites where H3K27ac was decreased (Fbw7$^{-/-}$: 6.9%, Fbw7$^{R/+}$: 4.3%) (*Figure 2C*, *Figure 2—source data 1*). Most nondifferential H3K27ac sites (those unaffected by Fbw7 status) were promoter-proximal, while loci with differential H3K27ac in either Fbw7$^{R/+}$ or Fbw7$^{-/-}$ cells fell mostly within introns or intergenic regions (p<0.0001, Fisher test) (*Figure 2D*, *Figure 2—figure supplement 2*). To determine whether these differential loci result from the altered binding of known Fbw7 substrates, we performed motif discovery analysis on the central 100-bp sequence of each peak. Strikingly, the AP-1 motif, which is bound by the Jun family, was found in 32% (p<1.8e−5) of the H3K27ac sites upregulated in Fbw7$^{-/-}$ cells (*Figure 2E*, *Figure 2—figure supplement 3A*). The AP-1 motif was also enriched in differential H3K27ac sites that were decreased in Fbw7$^{-/-}$ cells, as well as in differential H3K27ac sites in Fbw7$^{R/+}$ cells. In contrast, the AP-1 site was not enriched in H3K27ac sites that were unaffected by either Fbw7 mutation (*Figure 2—figure supplement 3B*). The AP-1 motif enrichment in these differential sites suggests that Fbw7-dependent Jun regulation may account, in part, for these changes in active chromatin.

## Fbw7 preferentially regulates Jun and Myc occupancy at distal regulatory regions

Fbw7 targets some TF-substrates while they are bound to DNA (*Fryer et al., 2004*; *Punga et al., 2006*). We thus speculated that substrates may recruit Fbw7 to chromatin and examined Fbw7-chromatin association in Hct116 cells with endogenous heterozygous (Fbw7$^{R/+}$) or homozygous

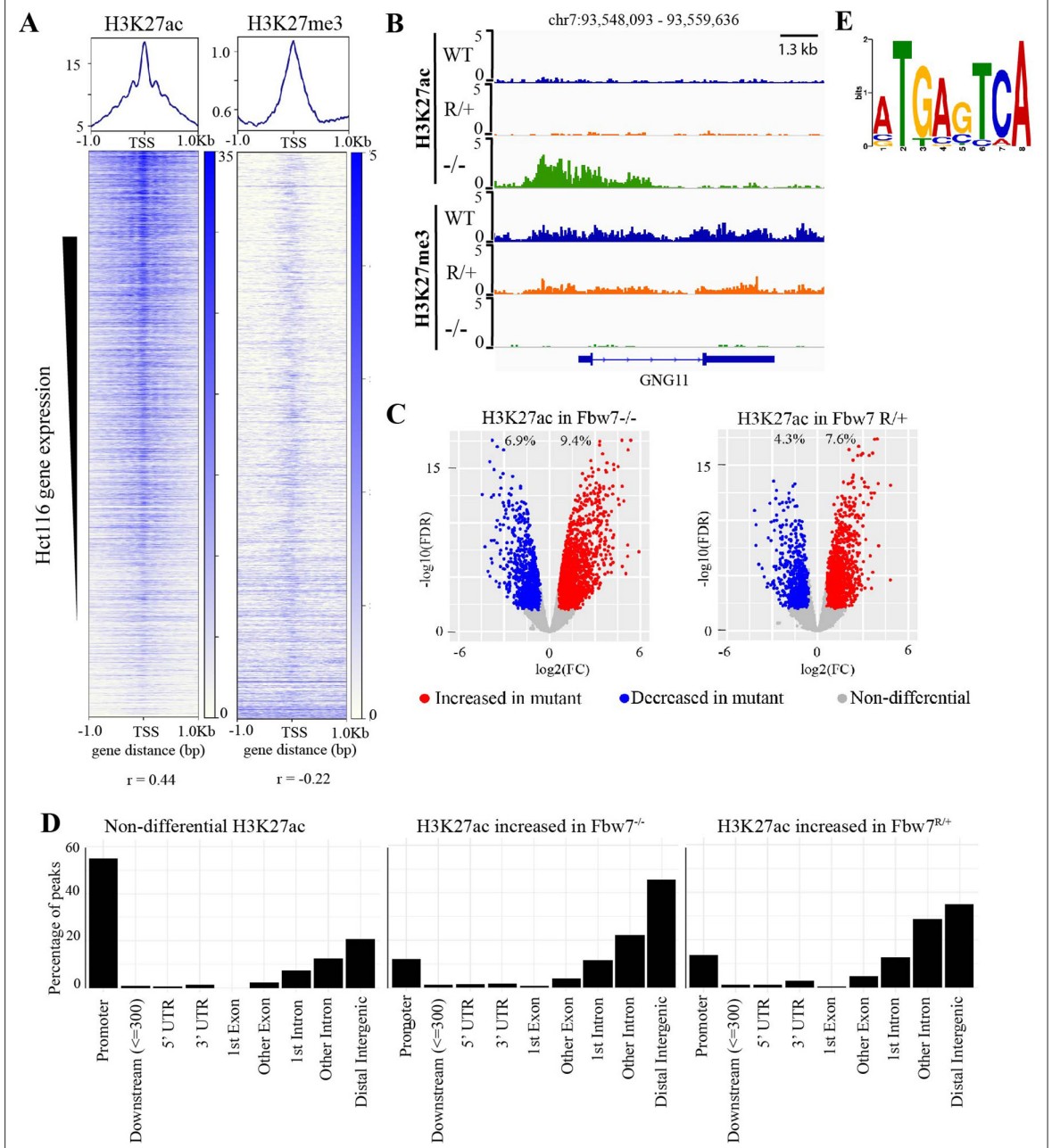

**Figure 2.** Differential H3K27ac signal in Hct116 Fbw7 mutant cells reveal genomic sites targeted by Fbw7. (**A**) Heatmaps showing the correlation between CUT&RUN profiles of H3K27ac and H3K27me3, and RNA-Seq in Hct116 WT cells. (**B**) Genome browser view of H3K27ac and H3K27me3 signal from Hct116 WT, Fbw7$^{R/+}$, and Fbw7$^{-/-}$ cells at a representative gene. (**C**) Peaks with increased (red) or decreased (blue) H3K27ac signal in Hct116 Fbw7$^{-/-}$ and Fbw7$^{R/+}$ cells compared to WT cells. Differential sites indicated as a percent of total H3K27ac peaks in Hct116 WT cells. (**D**) Percentage of H3K27ac peaks located within different gene regions. (**E**) Sequence logo for AP-1 motif enriched in H3K27ac peaks increased in Fbw7$^{-/-}$ cells (E value=1.6e−3). See *Figure 2—figure supplement 1*, *Figure 2—figure supplement 2*, *Figure 2—source data 1* and *Figure 2—source data 2*. *Figure 2—figure supplement 3* has the complete MEME output and details on the FIMO analysis. CUT&RUN, cleavage under target and release using nuclease; WT, wild-type.

The online version of this article includes the following source data and figure supplement(s) for figure 2:

**Source data 1.** H3K27ac differential sites.

**Source data 2.** Summary of CUT&RUN differential sites.

**Figure supplement 1.** Hierarchically clustered correlation matrix of H3K27ac CUT&RUN profiles in Hct116 cells.

**Figure supplement 2.** Percentage of peaks with decreased H3K27ac signal located within different gene features.

**Figure supplement 3.** Complete output of the MEME-ChIP analysis on H3K27ac differential sites.

(Fbw7$^{R/R}$) mutations. Fbw7 was found in both the chromatin and soluble fractions of WT cell lysates, but exclusively in the soluble fraction in Fbw7$^{R/R}$ cells (*Figure 3A*). The only known consequence of Fbw7$^R$ mutations is to prevent substrate binding; hence, the loss of chromatin-bound Fbw7 in Fbw7$^{R/R}$ cells suggests that substrates recruit Fbw7 to chromatin. Proteasome inhibition prevents substrate degradation and stabilizes Fbw7-substrate complexes. Accordingly, proteasome inhibition with bortezomib further shifted Fbw7 to the chromatin fraction (*Figure 3—figure supplement 1*), further supporting the hypothesis that substrate binding underlies Fbw7 chromatin association.

We next focused on two TF substrates: (1) Jun, because of the highly enriched AP-1 motifs in the Fbw7-dependent active chromatin, and (2) Myc, due to its prominent roles in Fbw7-associated cancer (*Davis et al., 2014*). Myc deregulation in Fbw7-associated cancers typically occurs through either Fbw7 or Myc-phosphodegron mutations (*Davis et al., 2014*; *Welcker et al., 2004*; *Yada et al., 2004*; *Yumimoto and Nakayama, 2020*). CPD phosphorylation and Myc ubiquitylation also modulate Myc transcriptional activity (*Endres et al., 2021*; *Gupta et al., 1993*; *Hemann et al., 2005*; *Jaenicke et al., 2016*; *Thomas and Tansey, 2011*). Myc is stabilized in Fbw7$^{-/-}$ Hct116 cells, but its steady-state abundance is less impacted due to negative autoregulation of Myc transcription (*Grim et al., 2008*). Myc is partially stabilized in Hct116 Fbw7$^{R/+}$ cells and Fbw7ΔD cells (in which endogenous Fbw7 dimerization is prevented) (*Davis et al., 2018*; *Welcker et al., 2013*; *Welcker et al., 2022*). These data implicate Fbw7 dimers in Myc turnover, which is mediated through two Myc degrons that concertedly bind Fbw7 dimers (*Welcker et al., 2022*). Fbw7 targets Jun for degradation after multisite phosphorylation in two identified degrons (*Csizmok et al., 2018*; *Nateri et al., 2004*; *Wei et al., 2005*). While we find the minimal impact of Fbw7 mutations on steady-state Jun abundance (*Davis et al., 2018*), several factors have been described that regulate Jun degradation by Fbw7 in Hct116 cells, including Rack1 (*Zhang et al., 2012*), BLM (*Priyadarshini et al., 2018*), and Usp28 (*Diefenbacher et al., 2014*).

We profiled genome-wide Jun and Myc occupancy (*Figure 3—figure supplement 2A*) to determine the extent that they are deregulated by Fbw7 mutations. As expected, Jun-binding and Myc-binding site motifs were highly enriched in the respective data sets (*Figure 3—figure supplement 2B*). Differential binding analyses of the Jun and Myc peaks demonstrated that 5.3% and 3.8% of the Jun sites and 2.2% and 3.3% of the Myc sites exhibited differential occupancy in Fbw7$^{-/-}$ and Fbw7$^{R/+}$ cells, respectively (*Figure 3B*, *Figure 2—source data 2*). Fbw7 mutations thus altered Myc and Jun occupancy at specific loci, rather than causing a global increase.

Like H3K27ac, most nondifferential Myc binding sites were promoter-proximal, whereas sites with increased Myc occupancy in Fbw7 mutant cells fell within introns and intergenic regions (p<0.001, Fisher test) (*Figure 3C*, *Figure 3—figure supplement 3*). Compared with Myc, a smaller proportion of the total Jun sites in WT-Hct116 cells were promoter-proximal, but again the sites with differential occupancy in Fbw7 mutant cells were heavily biased to intronic and intragenic regions (p<0.0001, Fisher test) (*Figure 3C*). The differential sites in introns and intergenic loci were enriched for H3K27ac and H3K4me1, indicating that they may function within distal regulatory elements, such as enhancers (*Figure 3D*).

To study the functional significance of Fbw7-dependent changes in Jun and Myc binding, we examined the expression of genes that could be linked to the differential Jun or Myc sites that fell within gene bodies or 10-kb upstream of TSS (*Figure 3E*). Approximately 39% of genes with increased promoter-proximal Jun occupancy and 46% of genes with decreased promoter-proximal Jun occupancy in Fbw7$^{-/-}$ cells exhibited corresponding increases or decreases in RNA expression (*Figure 3F*). Of note, many of the genes with promoter-proximal Jun or Myc sites were either not captured by RNA-Seq or had expression changes that were not statistically significant. Similar associations were seen with Myc differential sites, although fewer could be linked with transcripts (*Figure 3—figure supplement 4*). Overall, the differential sites that could be linked with associated genes showed good concordance between the changes in TF occupancy and RNA expression.

## Fbw7$^{-/-}$ and Fbw7$^{R/+}$ mutation-specific consequences

We next examined how Jun occupancy is differentially affected by Fbw7$^{R/+}$ and Fbw7$^{-/-}$ mutations. Importantly, many differential Jun sites were common to both mutant cell lines: 48% of differential Jun sites in Fbw7$^{R/+}$ (252/530; p<0.0001, Fisher test) and 35% of differential Jun sites in Fbw7$^{-/-}$ (252/715) (*Figure 4A*). Representative Jun peaks that are increased in Fbw7$^{-/-}$ and/or Fbw7$^{R/+}$ are shown in *Figure 4B*: (a) Jun occupancy at *KCNQ5* intronic sites was increased only in Fbw7$^{-/-}$ cells; (b) in *ITGA2*,

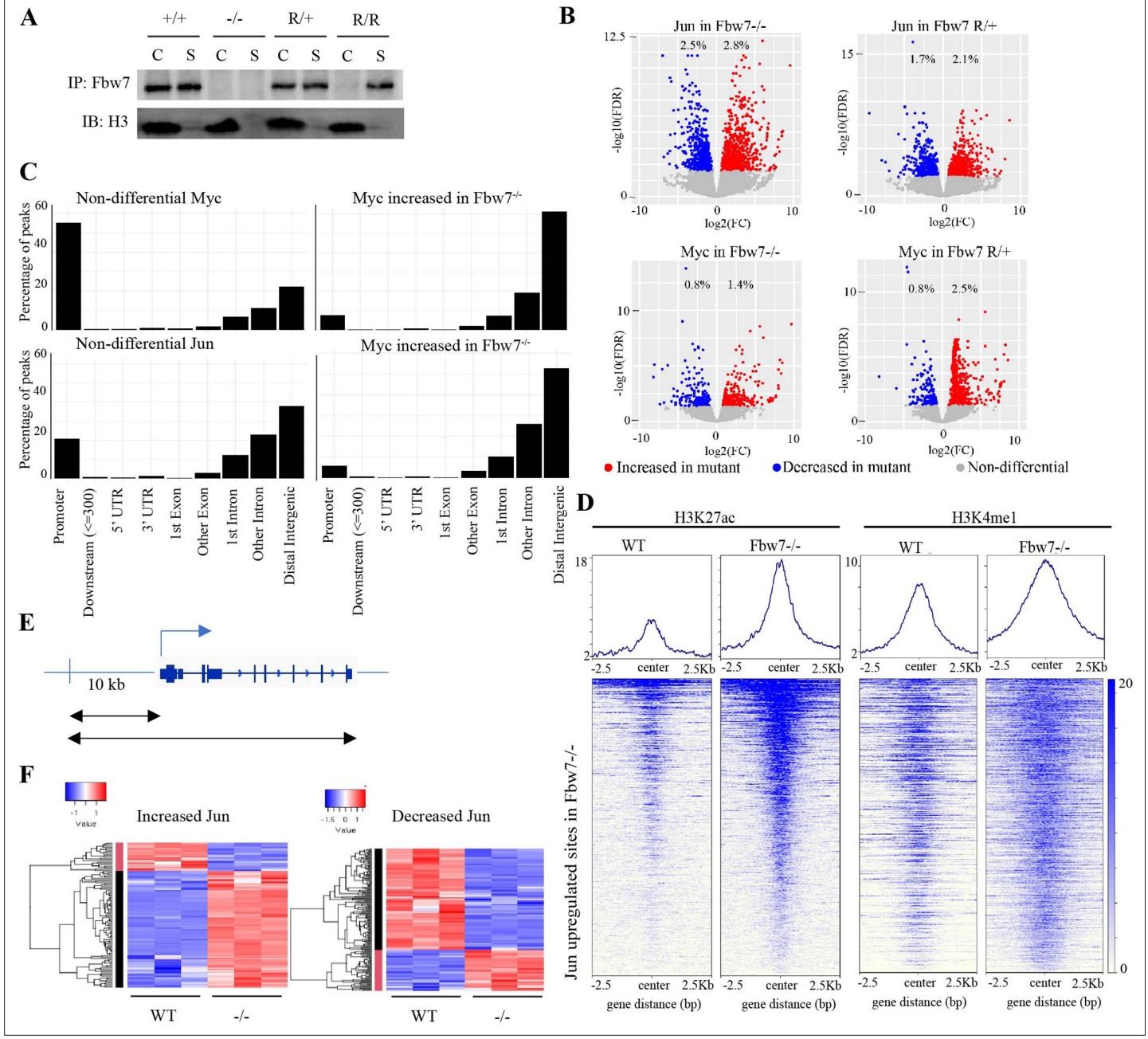

**Figure 3.** Fbw7 preferentially regulates Jun and Myc DNA occupancy at distal regulatory regions. (**A**) Fbw7 abundance in chromatin (C) and soluble (S) fractions from Hct116 WT, Fbw7$^{R/+}$, and Fbw7$^{R/R}$ cells. Histone H3 was detected in chromatin fractions. (**B**) Increased (red) and decreased (blue) Jun and Myc sites in Hct116 Fbw7$^{-/-}$ and Fbw7$^{R/+}$ cells compared to WT. (**C**) Nondifferential and differential Jun and Myc peaks located within gene features. (**D**) H3K27ac and H3K4me1 CUT&RUN signal from Hct116 WT and Fbw7$^{-/-}$ cells mapped on genomic sites that have increased Jun occupancy in Fbw7$^{-/-}$ cells. (**E**) Schema depicting the filtering criteria applied to the annotated differential sites to select gene proximal sites. (**F**) Transcription of genes with increased or decreased Jun bound at a gene proximal site. (Each row is a gene and three replicates each from Hct116 WT and Fbw7$^{-/-}$ cells are shown. Replicates for each genotype were from a single clone, however from separately cultured samples.) See ***Figure 3—figure supplements 1–4*** and ***Figure 3—source data 1–2***. CUT&RUN, cleavage under target and release using nuclease; WT, wild-type.

The online version of this article includes the following source data and figure supplement(s) for figure 3:

**Source data 1.** Original western blots for **Figure 3A** and **Figure 3—figure supplement 1**.

**Source data 2.** Jun and Myc differential sites in Hct116 cells.

**Figure supplement 1.** Fbw7 abundance in chromatin (C) and soluble (S) fractions from Hct116 WT, Fbw7$^{R/+}$, and Fbw7$^{R/R}$ cells treated with and without Bortezomib.

*Figure 3 continued on next page*

*Figure 3 continued*

**Figure supplement 2.** Validation of Jun and Myc CUT&RUN profiles.

**Figure supplement 3.** Percentage of Myc and Jun peaks located at different gene features.

**Figure supplement 4.** Transcription of genes with differential promoter-proximal Myc and Jun occupancy in Fbw7 mutant cells.

Jun occupancy was increased in both Fbw7$^{-/-}$ and Fbw7$^{R/+}$, but to an intermediate level in Fbw7$^{R/+}$; and (c) in *MAML2*, Jun occupancy was increased in Fbw7$^{R/+}$ more highly than in Fbw7$^{-/-}$.

Many sites, such as *ITGA2* exhibited an intermediate impact of Fbw7$^{R/+}$ on Jun occupancy, as depicted by the heatmap in which the Jun sites from WT, Fbw7$^{R/+}$, and Fbw7$^{-/-}$ cells were mapped on all the sites with increased Jun occupancy in Fbw7$^{-/-}$ cells (*Figure 4C*). H3K27ac sites followed this same trend (*Figure 4B and D*). RNA-Seq data also showed that some genes were deregulated in both Fbw7$^{-/-}$ and Fbw7$^{R/+}$, but to an intermediate level in Fbw7$^{R/+}$ (clusters 3 and 4; *Figure 1B and C*). These intermediately affected binding sites and transcripts, in which Fbw7$^{-/-}$>Fbw7$^{R/+}$>WT, consistent with the notion that the Fbw7$^{R/+}$ mutation is less severe than complete Fbw7 loss. Other Jun differential sites were uniquely impacted by either mutation (*Figure 4A*), including a subset of sites that were most strongly impacted by Fbw7$^{R/+}$. RNA-Seq data showed that genes in clusters 5 and 6 were deregulated most strongly in Fbw7$^{R/+}$ (*Figure 1B and C*). In summary, we identified differential Jun sites that are uniquely affected by each Fbw7 mutation type and others that were shared between the two mutant cell lines. Because the Fbw7$^{-/-}$ and Fbw7$^{R/+}$ cells were derived independently, the shared loci impacted by both mutations with respect to Jun occupancy, H3K27ac, and mRNA expression support the conclusion that these findings are attributable to Fbw7 status, rather than factors such as clonal variation.

## Fbw7 coordinately regulates Jun and Myc at co-occupied loci

Because Myc and Jun are oncogenic TFs with activities in shared pathways, we examined if they were coregulated at shared sites. Approximately 20% of the Myc and Jun binding sites overlapped in Hct116 WT cells (*Figure 4—figure supplement 1A*, p<0.0001, Fisher test). Jun and Myc exhibited strikingly coordinate regulation by Fbw7 at these co-occupied differential loci. We identified 78 sites in which both Jun and Myc occupancy were increased in Fbw7$^{-/-}$ cells and 53 sites where both Jun and Myc were decreased in Fbw7$^{-/-}$ cells (*Figure 4E and F*). In contrast, no sites with discordant changes in Jun and Myc occupancy (e.g., increased Jun but decreased Myc) were found (*Figure 4G and H*). We found similar concordance in coregulated Jun and Myc sites in Fbw7$^{R/+}$ cells (*Figure 4—figure supplement 1B and C*). We chose one of these coregulated loci, upstream of the *CIITA*, for further study.

## Jun and Myc coregulation by Fbw7 controls MHC Class II gene expression

Unlike MHC Class I genes, which are expressed in all cells, MHC Class II genes are normally expressed only in specific immune cells, where their expression is controlled by the Class II Major Histocompatibility Transactivator protein, or CIITA (*Masternak et al., 2000*; *Ting and Trowsdale, 2002*). CIITA and MHC Class II genes were upregulated in Fbw7$^{-/-}$ cells (*Figure 1C*). The *CIITA* gene contains four promoters (hereafter referred to as PI– PIV) that specify four transcripts with distinct first exons (*Muhlethaler-Mottet et al., 1997*). While CIITA isoform III is constitutively expressed in antigen-presenting cells, isoform IV is inducible by cytokines in nonhematopoietic cells (*van der Stoep et al., 2007*). The PIII Upstream Regulatory Region (PURR) is located 6-kb upstream of PIII and consists of regulatory sites for both constitutive and IFNγ-induced CIITA expression (*Deffrennes et al., 2001*; *van der Stoep et al., 2007*), as well as an AP-1 site (*Martins et al., 2007*). Both Myc and Jun bound to these upstream regulatory elements (PURR and an element 14-kb upstream of CIITA-PIII) and their occupancy were increased in Fbw7$^{-/-}$ cells (*Figure 5A*). Jun and Myc occupancy were also increased at these sites in Fbw7$^{R/+}$ cells, but to a lesser extent. H3K27ac and H3K4me1 were increased at these sites in Fbw7$^{-/-}$ cells, which is indicative of active transcription.

RNA-Seq revealed increased CIITA mRNA expression in Fbw7$^{-/-}$ cells (*Figure 5B*). Isoform-specific primers demonstrated that the pIII isoform is elevated in Fbw7$^{-/-}$ cells, but that the pIV isoform is not expressed (*Figure 5D*). Raji cells are shown as a control cell that expresses both CIITA isoforms. The upregulated CIITA expression in Fbw7$^{-/-}$ cells is functionally significant and caused increased

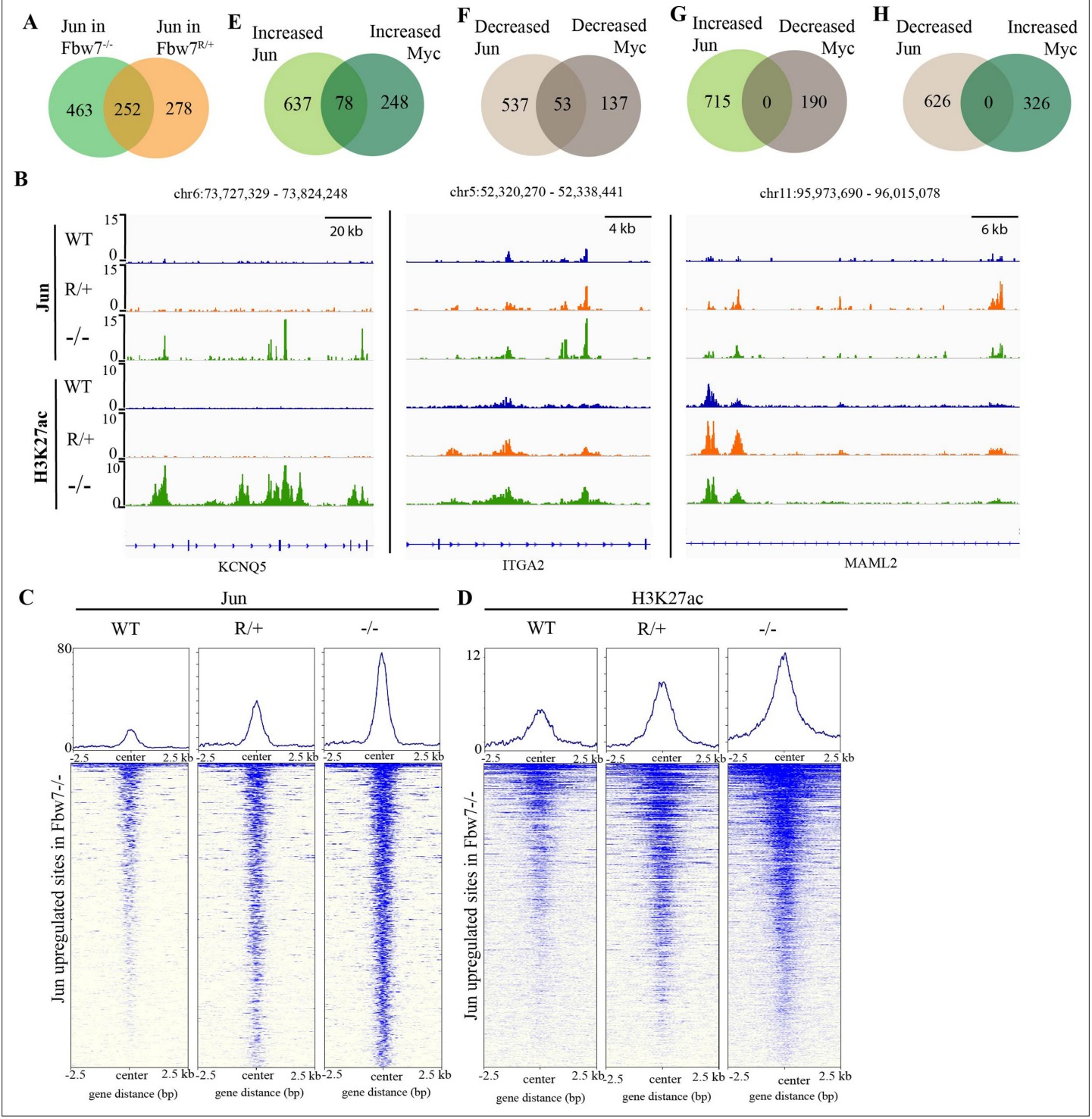

**Figure 4.** Fbw7 exhibits mutation-type specific regulation and coordinate regulation of multiple TFs. (**A**) The overlap between peaks with increased Jun in Fbw7$^{-/-}$ and Fbw7$^{R/+}$ cells. (**B**) Genome browser view of Jun and H3K27ac occupancy in Hct116 WT, Fbw7$^{-/-}$, and Fbw7$^{R/+}$ cells at representative loci. Black arrows point to peaks with increased signal uniquely in Fbw7$^{-/-}$ (*KCNQ5*), in both Fbw7$^{-/-}$ and Fbw7$^{R/+}$ (intermediate level in Fbw7$^{R/+}$) (*ITGA2*) and increased in Fbw7$^{R/+}$ than in Fbw7$^{-/-}$ (*MAML2*). (**C, D**) Heatmap of Jun and H3K27ac signal from each cell type mapped on sites with increased Jun in Fbw7$^{-/-}$ cells. (**E–H**) (**E**) the overlap between peaks with increased Jun and Myc, (**F**) decreased Jun and Myc, (**G**) increased Jun and decreased Myc, and (**H**) decreased Jun and increased Myc in Fbw7$^{-/-}$ cells. See *Figure 4—figure supplement 1*. TF, transcription factor; WT, wild-type.

The online version of this article includes the following figure supplement(s) for figure 4:

**Figure supplement 1.** Comparison between Jun and Myc peaks in Hct116 cells.

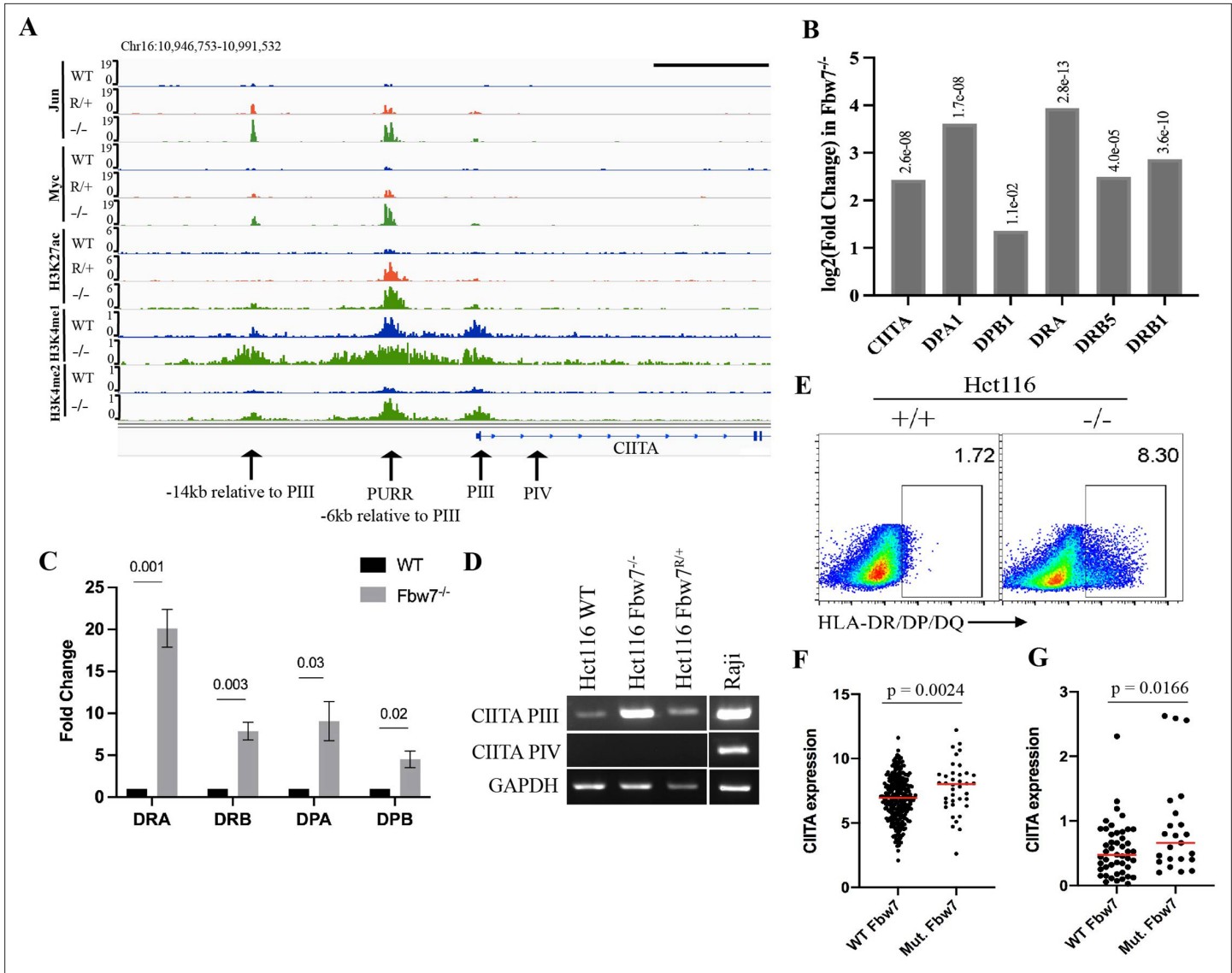

**Figure 5.** Fbw7 regulates the expression of MHC Class II genes. (**A**) Genome browser view of TFs and histone modification marks enriched at the promoter and regulatory sites upstream of *CIITA* gene. Arrows point to (from right to left): PIV (promoter of isoform IV); PIII (promoter of isoform III); PURR (PIII Upstream Regulatory Regions)—a known regulatory site 6-kb upstream of PIII; and a known regulatory site 14-kb upstream to PIII. Black scale bar=10 kb. (**B**) Expression fold change of CIITA and MHC Class II genes in Hct116 Fbw7⁻/⁻ with respect to WT cells. FDR values are indicated on top of each bar. n=3. (**C**) Quantitative RT-PCR analysis of MHC Class II (HLA-DRA, HLA-DRB, HLA-DPA, and HLA-DPB) expression in Hct116 Fbw7⁻/⁻ cells. Mean fold change in Fbw7⁻/⁻ cells with respect to WT cells. Error bars=SEM, n=3. (**D**) CIITA isoforms III and IV amplified using isoform specific primers in Hct116 and Raji cells. (**E**) Flow cytometry analysis of HLA-DR/DP/DQ protein expression in Hct116 cells. (**F**) CIITA expression in primary cancer samples from TCGA COADREAD data sets that have WT Fbw7 (n=297) and mutated Fbw7 (n=43). (**G**) CIITA expression in colon and rectal cancer cell lines with WT Fbw7 (n=47) and mutated Fbw7 (n=23). Data collected from DepMap portal. See *Figure 5—figure supplement 1*, *Figure 5—figure supplement 2* and *Figure 5—source data 1–4*. TF, transcription factor; WT, wild-type.

The online version of this article includes the following source data and figure supplement(s) for figure 5:

**Source data 1.** Original gels for *Figure 5C*.

**Source data 2.** TCGA COADREAD data used for *Figure 5D*.

**Source data 3.** Colorectal cancer cell line data from DepMap used for *Figure 5E*.

**Source data 4.** Quantitative RT-PCR analysis of MHC Class II genes in Hct116 cells.

**Figure supplement 1.** Expression fold change of MHC Class I genes in Hct116 Fbw7⁻/⁻ with respect to WT cells.

**Figure supplement 2.** MHC Class II protein expression in Hct116 cells.

**Figure supplement 2—source data 1.** Original western blots for *Figure 5—figure supplement 2*.

expression of MHC Class II genes (HLA-DPA, HLA-DPB, HLA-DRB, and HLA-DRA), as shown by both RNA-Seq and by qPCR using primers that detect MHC Class II mRNAs (*Figure 5B and C*). In contrast, MHC Class I genes were not DE in Fbw7 mutant cells (*Figure 5—figure supplement 1*). Flow cytometry revealed increased and heterogeneous MHC Class II protein surface expression in Fbw7$^{-/-}$ cells and immunoblotting detected increased protein expression (*Figure 5E*, *Figure 5—figure supplement 2*). The basis for this heterogeneity in protein expression is not presently understood.

We also analyzed CIITA expression in primary CRCs in TCGA data sets, which was increased in Fbw7 mutant cancers compared with Fbw7 WT tumors (*Figure 5F*). Because these primary tumors contain immune infiltrates, the increased CIITA expression could result from CIITA expression in either tumor cells or immune cells. We thus analyzed CRC cell lines in the Cancer Cell Line Encyclopedia, which also revealed elevated CIITA expression in Fbw7 mutant cell lines (*Figure 5G*; *Ghandi et al., 2019*). Because many of the TCGA and CCLE CRC specimens contain FBW7$^{R/+}$ mutations, these analyses underrepresent MHC Class II overexpression in Fbw7$^{-/-}$ CRCs. These data support the idea that Fbw7 regulates CIITA expression in CRC, likely due to coregulation of Myc and Jun at the PIII upstream regulatory site.

## Acute Fbw7 loss in NSCs recapitulate findings from Hct116 cells

Because the Fbw7 mutations in the Hct116 cell panel were stably engineered into cells that are transformed and clonal, we examined the generalizability of these results by determining how acute Fbw7 deletion in non-transformed cells impact RNA expression and Jun occupancy. We studied U5 NSCs (*Bressan et al., 2017*), which represent a cell type in which Fbw7 has important developmental and oncogenic activities, and which displays Fbw7-mediated Jun regulation (*Hoeck et al., 2010*). We used a high efficiency CRISPR/nucleofection protocol to inactivate Fbw7 in two independent experiments without the need for any selection (*Figure 6—figure supplement 1A*; *Hoellerbauer et al., 2020b*; *Hoellerbauer et al., 2020a*). Fbw7 was efficiently deleted in U5 NSCs, which modestly impacted S-phase entry and doubling time (*Figure 6—figure supplement 2*). Analogous to the Hct116 cell panel, ~9% of protein-coding genes were DE in Fbw7$^{-/-}$ cells compared with WT-U5 NSCs (*Figure 6A*).

We identified enriched gene sets in the DE transcripts in Fbw7$^{-/-}$ cells (*Figure 6B* and *Figure 6—source data 2*). Surprisingly, many of these pathways were the same as those identified in the Hct116 cells, which included EMT- and ECM-related gene sets, the p53 pathway, and MHC Class II genes. Gene sets associated with KLF-binding sites were also highly enriched in upregulated transcripts in Fbw7$^{-/-}$ NSCs, but not in the downregulated transcripts. While the significance of these broadly defined gene sets common to both NSC and Hct116 cells will require both validation and further study, their conservation between these disparate cell types raises the possibility that they represent core normal and neoplastic Fbw7 functions (see Discussion). The most significant enriched genet set in downregulated differential genes was nervous system development, consistent with Fbw7's role in regulating NSC differentiation. We also compared the specific Fbw7-dependent differential genes in Fbw7$^{-/-}$ NSCs with those in Fbw7$^{-/-}$ Hct116 cells (*Figure 6—figure supplement 6*). The gene clusters exhibiting the greatest Fbw7-dependence in each heat map were relatively devoid of genes that were expressed in both cell types, suggesting that the most highly Fbw7 dependent genes were those with cell-type specific expression.

Jun occupancy in WT and Fbw7$^{-/-}$ NSCs closely mirrored that seen in the Hct116 cell panel (*Figure 6—figure supplement 1*) in that: (1) only a subset of the Jun binding sites displayed differential occupancy after Fbw7 deletion (8.3% increased and 4.1% decreased) sites, and (2) most of the differentially regulated Jun sites occurred in introns and intergenic regions (p<0.0001, Fisher test) (*Figure 6C and D*, *Figure 6—figure supplement 3*). Thus, while most of the specific loci impacted by Fbw7 loss in the Hct116 cells NSCs differed, the scope of Fbw7's impact on Jun was quite similar. Upregulation of CIITA and MHC Class II expression (*Figure 6E*) after Fbw7 loss was also seen in both Hct116 cells and NSCs. Jun and Myc were bound at regulatory regions upstream of CIITA in NSCs, but only Myc occupancy was increased in Fbw7$^{-/-}$ NSCs (*Figure 6F*). Unlike Hct116 cells, NSC cells express a basal level of CIITA, perhaps consistent with constitutive Jun occupancy upstream of CIITA (*Figure 6—figure supplement 4*). RNA-Seq and RT-PCR revealed the increased MHC Class II gene expression after Fbw7 deletion, which also increased MHC Class II protein expression (*Figure 6—figure supplement 5*).

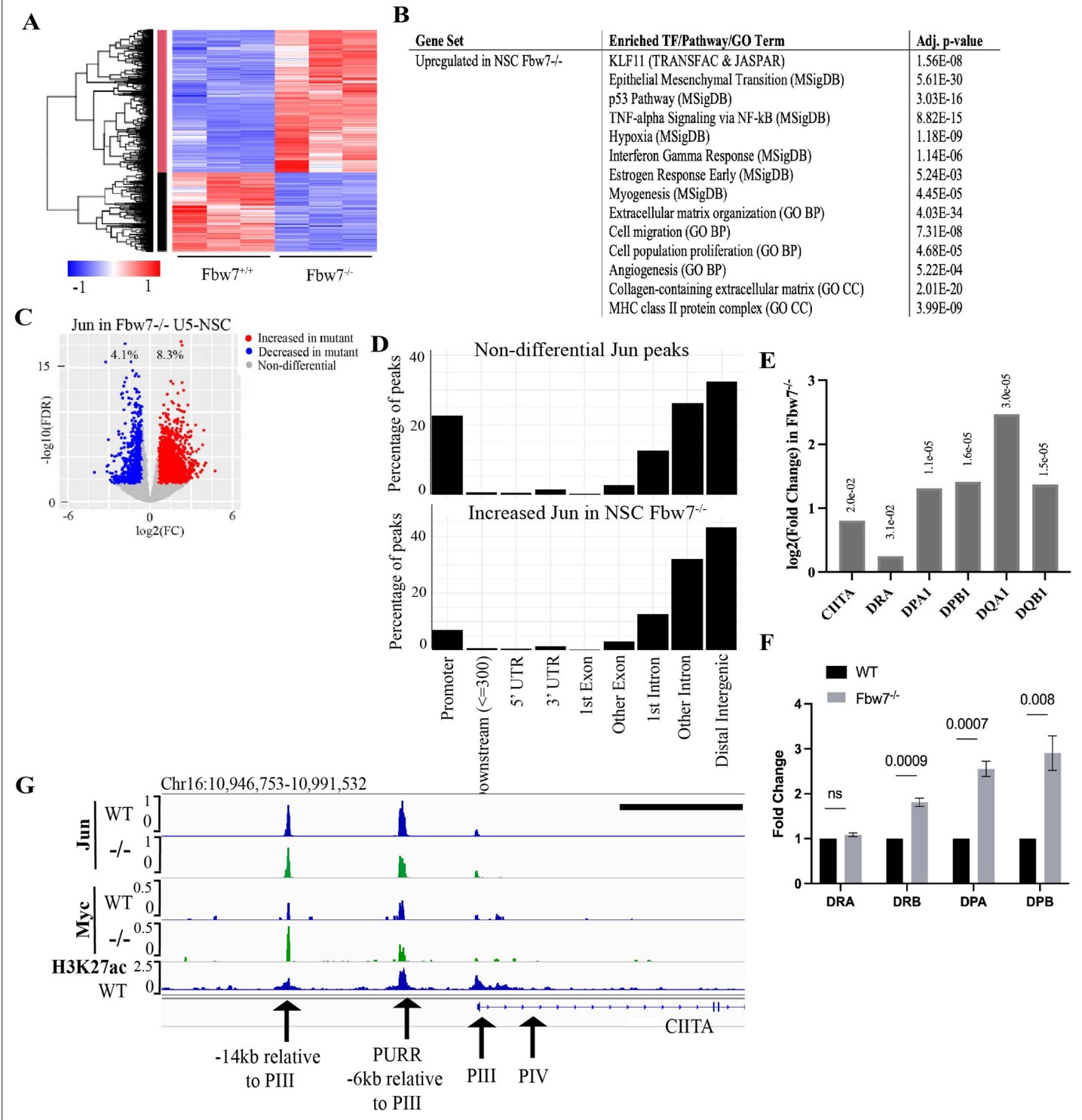

**Figure 6.** Transcriptional consequences of loss of Fbw7 in neural stem cells. (**A**) Clustering analysis separates differentially expressed protein-coding genes in NSCs into two groups. Heatmap shows the intensity of expression of each gene (y-axis) for three replicates per cell type (x-axis). Three replicates were from two independently engineered cell samples. (**B**) TFs, pathways, and GO terms enriched in upregulated genes in Fbw7$^{-/-}$ NSCs. (**C**) Sites with increased (red) and decreased (blue) Jun in Fbw7$^{-/-}$ NSCs compared to WT. (**D**) Nondifferential and differential Jun peaks located within each gene feature. (**E**) Fold change of CIITA and MHC Class II genes in Fbw7$^{-/-}$ NSCs compared to WT. FDR values are given at the top of each bar. n=3. (**F**) Quantitative RT-PCR analysis of MHC Class II (HLA-DRA, HLA-DRB, HLA-DPA, and HLA-DPB) expression in Fbw7$^{-/-}$ NSCs. Mean fold change in Fbw7$^{-/-}$ cells with respect to WT cells. Error bars=SEM, n=3. (**G**) Genome browser view of Myc, Jun, and H3K27ac occupancy on CIITA regulatory regions in WT and Fbw7$^{-/-}$ NSCs. Black scale bar=8 kb. See *Figure 6—figure supplements 1–3* and *Figure 6—source data 1–4*, *Figure 6—figure*

*Figure 6 continued on next page*

*Figure 6 continued*

*supplement 1—source data 1*, *Figure 6—figure supplement 3—source data 1*, *Figure 6—figure supplement 5—source data 1*. GO, gene ontology; TF, transcription factor; WT, wild-type.

The online version of this article includes the following source data and figure supplement(s) for figure 6:

**Source data 1.** Differential expression analysis of U5-NSC RNA-Seq.

**Source data 2.** Enrichr output for U5-NSC differentially expressed genes.

**Source data 3.** Jun differential sites in U5-NSCs.

**Source data 4.** Quantitative RT-PCR analysis of MHC Class II genes in NSCs.

**Figure supplement 1.** Validation of U5-NSC Fbw7$^{-/-}$ generation and CUT&RUN Jun signal.

**Figure supplement 1—source data 1.** Confirming loss of Fbw7 in U5-NSC Fbw7$^{-/-}$ cells.

**Figure supplement 2.** Proliferation of Fbw7 mutant U5 NSCs.

**Figure supplement 3.** Percentage of peaks with decreased Jun in U5 NSC Fbw7$^{-/-}$ within different gene regions.

**Figure supplement 3—source data 1.** Original gels for *Figure 6—figure supplement 3*.

**Figure supplement 4.** CIITA isoform III amplified using isoform specific primers in U5 NSCs.

**Figure supplement 5.** MHC Class II protein expression in Hct116 cells.

**Figure supplement 5—source data 1.** Original western blots for *Figure 6—figure supplement 5*.

**Figure supplement 6.** Comparison of differentially expressed (DE) genes in Hct116 and NSCs.

## Discussion

Our overarching goal was to understand the global transcriptional consequences of oncogenic Fbw7 mutations. Overall, Fbw7 loss affected ~10% of expressed genes, which could result from Fbw7 substrates that are sequence-specific TFs or global transcriptional regulators, such as the Mediator complex (*Davis et al., 2013*). However, only relatively small subsets of the mapped H3K27ac and Jun/ Myc binding sites were affected by Fbw7 status. What might account for this specificity? Genomic location appears to be one factor, since most differential sites fell within distal regulatory elements. By targeting an enhancer rather than individual promoters, Fbw7 might cooperatively regulate multiple genes via a single regulatory region. However, we do not presently understand why these loci are particularly dependent on Fbw7 function. One possibility is that TF phosphorylation may be restricted to just the differential sites, thereby limiting where Fbw7 functions. If so, we might expect to find Fbw7 bound to these sites, as supported by our data implicating Fbw7 recruitment to chromatin by TFs (*Figure 3—figure supplement 1*). However, we have not yet successfully mapped endogenous Fbw7 to specific chromatin sites.

Only a minority of Jun and Myc binding sites were differentially regulated by Fbw7, but there was substantial overlap and uniform coregulation of the two TFs. That is, in every case where they overlapped, both Jun and Myc occupancy were coordinately increased or decreased by Fbw7 loss. We speculate that Fbw7 regulates transcription at sites such as CIITA through the concerted targeting of both Jun and Myc. The expected outcome of Fbw7 loss is substrate accumulation, and most differential sites had increased occupancy in Fbw7 mutant cells. However, we also found differential sites with decreased Jun or Myc occupancy and correspondingly less mRNA expression in Fbw7 mutant cells. The mechanisms through which Fbw7 loss decreases TF occupancy remain to be elucidated and could involve the many TFs and associated proteins that are targeted by Fbw7. While Myc ubiquitylation impacts transcriptional elongation (*Jaenicke et al., 2016*), our data do not presently provide insights into how this may contribute to the observed genomic distribution of Fbw7-dependent loci. We examined differential Myc binding sites for specific Myc cellular functions that might be targeted by Fbw7, such as protein synthesis and metabolism. We did not find any evidence for this regulation, but the low number of differential Myc-binding sites that we could link to genes may have limited this analysis.

Fbw7$^{R/+}$ mutations may stabilize those substrates that require a fully functional Fbw7 dimer and the subset of transcripts impacted by Fbw7$^{R/+}$ could thus result from dimer-dependent Fbw7 functions (*Welcker et al., 2013*). If so, the reduced transcriptional consequences seen in Fbw7$^{R/+}$ cells (compared with Fbw7$^{-/-}$) may reflect the regulation of just those substrates impacted by heterozygous Fbw7 missense mutations. Moreover, the many genes and loci with intermediate levels of

deregulation in Fbw7$^{R/+}$ cells could be consistent with the 'just-enough' model of Fbw7$^{R/+}$ function. We did not find enriched gene sets that were unique to Fbw7$^{R/+}$ cells. Instead, most of the enriched gene sets in Fbw7$^{R/+}$ were also found in Fbw7$^{-/-}$ cells. While the biologic significance of these enriched gene sets requires experimental validation, they suggest that regulation of cancer-associated processes, like EMT, ECM, and cell migration, may be critical and shared consequences of Fbw7$^{-/-}$ and Fbw7$^{R/+}$ mutations. The Fbw7$^{R/+}$ and Fbw7$^{-/-}$ cells not only shared enriched pathways, but also specific genes assigned to these pathways. For example, 10/17 of the Fbw7$^{R/+}$ genes assigned to the MSigDB EMT pathway were also found in Fbw7$^{-/-}$ cells, as well as 11/15 Fbw7$^{R/+}$ genes in the MSigDb p53 pathways set.

Despite the many differences between our studies in Hct116 cells and NSCs (e.g., cell type, acute versus chronic Fbw7 loss, transformed versus non-transformed cells, and clonal versus non-clonal), the remarkably similar consequences of Fbw7 loss in both cases suggests that these features are fundamental properties of Fbw7's normal function and its mechanisms of tumor suppression. For example, the extent of Fbw7-dependent changes in transcription, H3K27ac marks, and Jun binding sites was highly conserved between the two cell types, as was the genomic distribution of the Fbw7-regulated differential sites. There was also extensive overlap between the enriched biological processes found in differential transcripts in Fbw7$^{-/-}$ NSCs and Fbw7$^{-/-}$ Hct116 cells. Pathways related to EMT and ECM were the most highly enriched pathways in NSCs, and they were found in both Fbw7 mutant Hct116 cell lines. Pathways related to EMT (e.g., ECM and cell migration) may thus represent core Fbw7 targets in cancer and development that are conserved across disparate cell types. Future studies will focus on identifying the Fbw7 substrate(s) responsible for these findings, which seem likely to reveal new insights into Fbw7 function. Another enriched pathway associated with Fbw7 loss in both NSCs and Hct116 cells involved genes with binding sites for the KLF proteins, a family of TFs with diverse roles in cell proliferation and differentiation. Several KLF proteins are Fbw7 substrates (KLF2/4/5/10/11); these proteins all share the same binding site consensus and function as either repressors and/or activators (*McConnell and Yang, 2010*). These data implicate KLFs as potential mediators of Fbw7-dependent transcription, and the absence of the KLF motif in downregulated genes may further implicate KLF proteins that are transcriptional activators. Finally, another shared consequence of Fbw7 loss in Hct116 cells and NSCs is slower proliferation, which is likely due to both homeostatic responses (e.g., p53) and intrinsic activities of Fbw7 substrates (e.g., cyclin E). It will thus be important to ascertain how proliferation may relate to the enriched gene sets associated with Fbw7 mutations.

Constitutive CIITA expression is normally confined to antigen-presenting cells and it was thus striking to find CIITA and MHC Class II genes expressed in Fbw7 mutated colon cancer cells and NSCs. Abnormal CIITA and MHC Class II expression occur in tumors, including CRCs (*Axelrod et al., 2019*; *Sconocchia et al., 2014*). Aberrant CIITA expression in melanomas results from the activation of both IFNγ-inducible and constitutive CIITA promoters, and deletion of the same AP-1 site that we found differentially occupied by Jun in Fbw7 mutant Hct116 cells compromised CIITA expression in melanoma cells (*van der Stoep et al., 2007*). The data support a model in which Fbw7 loss augments CIITA expression through increased Jun and Myc occupancy at these regulatory regions.

What are the implications of Fbw7-dependent CIITA expression in CRCs? Tumor cell-specific MHC Class II expression is generally associated with favorable prognosis, which may be due to increased tumor immunogenicity conferred by MHC Class II expression. Intriguingly, Fbw7 mutations and distant metastasis do not frequently co-occur in CRCs (*Muzny et al., 2012*). Fbw7 loss may thus confer better prognosis in CRCs, perhaps due to Fbw7-dependent MHC Class II upregulation. Accordingly, we found increased CIITA expression in TCGA colorectal tumors and CCLE CRC cell lines with Fbw7 mutations. We previously used machine learning to develop gene expression signatures that predicted Fbw7 mutations in TCGA tumors. While we focused on a metabolic gene signature in that study, we also found that a signature comprised of MHC Class II and other genes associated with immune cells was highly predictive of Fbw7 mutations in CRCs (*Davis et al., 2018* Supplemental Data set S02). While these analyses were correlative, considering our current finding that Fbw7 loss induces CIITA expression in Hct116 cells and NSCs, we speculate that Fbw7 mutations in CRCs lead to increased CIITA expression, increased immunogenicity, and better prognosis.

Other associations between Fbw7 and immune responses have also been reported. A recent study found that an Fbw7$^{R/+}$ mutation conferred resistance to PD-1 blockade through impaired dsRNA sensing and IFNγ signaling in a metastatic melanoma and a murine melanoma model (*Gstalder et al.,*

2020). In this case, Fbw7$^{R/+}$ decreased MHC Class I but not MHC Class II expression and caused a more aggressive phenotype associated with decreased immunogenicity. These discrepancies with our study, which show upregulation of MHC Class II and no impact on MHC Class I, may relate to the different tumor types and/or model systems. In mice, Fbw7 also regulates the tumor microenvironment through a non-tumor-cell-autonomous manner involving the expression of the CCL2 chemokine (*Yumimoto et al., 2015*). Further studies are thus needed to fully appreciate the pathologic and therapeutic implications of Fbw7-related tumor immunogenicity.

Finally, it is important to note the potential impact of clonal variation on our studies, which involved Hct116 cell clones. Several factors increase our confidence that our findings we describe are directly attributable to Fbw7 status, rather than clonal evolution. First, we found that a large proportion of differential loci and transcripts were shared between the two Fbw7 mutant Hct116 cell lines (*Figures 1C and 4A*), and many of these exhibited a gradient of deregulation that tracked with the severity of the mutations (Fbw7$^{-/-}$>Fbw7$^{R/+}$). Because these cells were independently derived, it is unlikely that these shared features arose stochastically through clonal evolution in each line. Next, we acutely deleted Fbw7 in the orthogonal and non-clonal NSCs to mitigate against this possibility, and the shared consequences of Fbw7 loss (proportion of differential sites and transcripts, intergenic location of differential sites, and shared biological processes) in these two disparate cell systems support the conclusion that these observations reflect shared and direct consequences of Fbw7 loss. Finally, we studied biological replicates of non-clonal NSCs in which Fbw7 was independently inactivated by high-efficiency nucleofection-based CRISPR/Cas9 editing, and the consistent results across these replicates indicate that the differential loci are unlikely to result from 'drift' phenomena. In future studies, we plan to adopt the strategy of using high-efficiency gene targeting to acutely delete Fbw7 in primary cells, to study other tissues and stem cells. These studies will also allow us to again assess any contributions of clonality.

In summary, these data establish a framework for understanding the global and context-specific effects of Fbw7 mutations across cell types and cancers, and how Fbw7 substrates may act synergistically to control gene expression.

# Materials and methods

**Key resources table**

| Reagent type (species) or resource | Designation | Source or reference | Identifiers | Additional information |
|---|---|---|---|---|
| Gene (*Homo sapiens*) | *FBXW7* | NCBI | Gene ID: 55294 | Also known as Fbw7 and FBXW7 |
| Gene (*H. sapiens*) | *JUN* | NCBI | Gene ID: 3725 | Also known as AP1, AP-1, cJUN, and c-Jun |
| Gene (*H. sapiens*) | *MYC* | NCBI | Gene ID: 4609 | Also known as MYC, c-Myc, cMyc |
| Cell line (*H. sapiens*) | Hct116 | From Dr. Julian Simon at Fred Hutch | HCT116 CCL-247 | Adult male colon cancer |
| Cell line (*H. sapiens*) | U5 human neural stem cells (NSCs) | Jackson Laboratory | B6.129P2Gpr37tm1Dgen/J | Primary cells |
| Antibody | Anti-H3K27ac Rabbit Monoclonal | Abcam | Cat#: ab45173 | C&R (1:100) |
| Antibody | Anti-H3K27me3 Rabbit Monoclonal (C36B11) | Cell Signaling Technology | Cat#: 9733A | C&R (1:100) |
| Antibody | Anti-cJun Rabbit Polyclonal (H-79) | Santa Cruz Biotechnology | Cat#: sc-1694 | C&R (1:25) |
| Antibody | Anti-Myc Rabbit Monoclonal (D3N8F) | Cell Signaling Technology | Cat#: 13987S | C&R (1:25) |
| Antibody | Anti-H3K4me1 Rabbit Polyclonal | Abcam | Cat#: ab8895 | C&R (1:100) |
| Antibody | Anti-H3K4me2 Rabbit Monoclonal (C64G9) | Cell Signaling Technology | Cat#: 9725 | C&R (1:100) |
| Antibody | Normal IgG (Rabbit) | Santa Cruz Biotechnology | sc-2027 | C&R (1:50) |
| Antibody | Anti-Fbw7 Rabbit Polyclonal | Bethyl | A301-720A | WB (1:1000) |

*Continued on next page*

*Continued*

| Reagent type (species) or resource | Designation | Source or reference | Identifiers | Additional information |
|---|---|---|---|---|
| Antibody | Anti-HLA DR Mouse Monoclonal TAL 1B5 | Abcam | ab20181 | WB (1:500) |
| Antibody | Anti-HLA DR+ DP + DQ Mouse Monoclonal CR3/43 | Abcam | ab7856 | WB (1:500) |
| Antibody | Anti-Mouse IgG peroxidase-linked secondary | Cytiva NA9311ML | Cat#: 45-000-679 | WB (1:10,000) |
| Antibody | Anti-Rabbit IgG peroxidase-linked secondary | Cytiva NA9341ML | Cat#: 45-000-682 | WB (1:10,000) |
| Antibody | Anti-γ Tubulin Mouse Monoclonal | Santa Cruz Biotechnology | sc-17787 | WB (1:1000) |
| Antibody | Anti-H3 Mouse Monoclonal (96C10) | Cell Signaling Technology | Cat#: 3638S | WB (1:1000) |
| Antibody | PE Anti-HLA DR+ DP + DQ Mouse Monoclonal WR18 | Abcam | ab23901 | FC (1:50) |
| Antibody | PE Anti-HLA DR Mouse Monoclonal | BioLegend | Cat#: 307605 | FC (1:50) |
| Commercial assay or kit | QIAGEN RNeasy Mini Kit | QIAGEN | Cat#: 74104 | |
| Commercial assay or kit | iScript Reverse Transcription Supermix | Bio-Rad | Cat#: 1708841 | |
| Commercial assay or kit | Platinum SYBR Green qPCR SuperMix-UDG with Rox | Invitrogen | Cat#: 11744100 | |
| Software, algorithm | Beckman Biomek FX liquid handling robot | Genomics and Bioinformatics Center at Fred Hutch | | Automated CUT&RUN |
| Software, algorithm | ChemiDoc Touch Imaging System OS 2.3.0.07 | Bio-Rad | | |

## RNA-Seq: RNA isolation, library preparation, sequencing, and data analysis

RNA was isolated using the QIAGEN RNeasy Mini Kit (Cat#: 74104) following the manufacturer's instructions. Three replicates per cell type were included for each replicate. Three replicates per cell condition were included in the experiments. Hct116 triplicates were obtained from the same genetically engineered clone; however, they were harvested from separately cultured plates. NSC replicates were generated with two independent CRISPR/Cas9 nucleofections and cultured separately for the three independent samples. RNA quality and integrity were determined (A260/280 1.8–2.1, A260/230>1.7, RIN≥9). Libraries were prepared by the Fred Hutch Genomics Center using the TruSeq RNA Samples Prep Kit v2 (Illumina Inc, San Diego, CA). Sequencing was performed on an Illumina HiSeq 2500 with 50-bp paired-end reads (PE50). RNA-Seq for U5-NSCs was an exception. Knockouts were generated separately on 2 different days and cells from separate nucleofection reactions were used as the three replicates, hence biological replicates. Libraries were prepared using TruSeq Stranded mRNA and sequencing was performed using an Illumina NovaSeq 6000 employing a paired-end 50 base read length (PE50).

Fastq files were filtered to exclude reads that did not pass Illumina's base call quality threshold. STAR v2.7.1 (*Dobin et al., 2013*) with two-pass mapping was used to align paired-end reads to human genome build hg19 and GENCODE gene annotation v31lift37 (https://www.gencodegenes.org/human/). FastQC 0.11.8 (https://www.bioinformatics.babraham.ac.uk/projects/fastqc/) and RSeQC 3.0.0 (*Wang et al., 2012*) were used for QC including insert fragment size, read quality, read duplication rates, gene body coverage, and read distribution in different genomic regions. FeatureCounts (*Liao et al., 2014*) in Subread 1.6.5 was used to quantify gene-level expression. For stranded libraries, only coding strand derived reads were counted. Bioconductor package edgeR 3.26.8 (*Robinson et al., 2010*) was used to detect differential gene expression between conditions. Genes with low expression were excluded by requiring at least one count per million in at least N samples (N is equal to one less than the number of samples in the smallest group). The filtered expression matrix was normalized by TMM method (*Robinson and Oshlack, 2010*) and subject to significance testing using generalized linear model and quasi-likelihood method. Genes were deemed DE if absolute fold changes were above 1.5 and FDRs were less than 0.05.

## Cleavage under target and release using nuclease

Three replicates per cell condition were included in the experiments. Hct116 triplicates were obtained from the same genetically engineered clone; however, they were harvested from separately cultured plates. NSC replicates were generated with two independent CRISPR/Cas9 nucleofections and cultured separately for the three independent samples. Manual or automated CUT&RUN was performed as previously described (*Janssens et al., 2018*; *Skene et al., 2018*; *Skene and Henikoff, 2017*). Briefly, cells were harvested using Accutase, counted and washed twice with Wash Buffer (20 mM HEPES pH 7.5, 150 mM NaCl, 0.5 mM spermidine, and one Roche Complete EDTA-free protein inhibitor tablet per 50 ml). Cells were bound to Concanavalin A-coated magnetic beads (20 µl per one million cells). Then cells were permeabilized with Dig Wash buffer (Wash Buffer with 0.05% digitonin) while being incubated with primary antibody overnight at 4°C. Cell-bead mixture was washed twice with Dig-Wash buffer and incubated with Protein A-MNase (pA-MN) for 1 hr at 4°C. After washing the mix with Dig Wash buffer twice, cells were placed on an ice-cold block and incubated with 2 mM $CaCl_2$ in Dig Wash buffer to activate pA-MN digestion. After the specific digestion period, the reaction was inhibited with 2× Stop Buffer (340 mM NaCl, 20 mM EDTA, 4 mM EGTA, 0.05% digitonin, 0.05% mg/ml glycogen, 5 µg/ml RNase, and 2 pg/ml heterologous spike-in DNA). The samples were incubated at 37°C for 30 min to release the digested DNA fragments into the supernatant. The supernatant was collected and libraries were prepared as previously explained (*Janssens et al., 2018*). Paired-end 25 base read length (PE25) sequencing was performed using an Illumina HiSeq 2500 platform at Fred Hutch Genomics Shared Resources.

## Deviations from the above described protocol

1.  Automated CUT&RUN: manual preparation included harvesting cells, counting, washing, permeabilizing, and antibody addition. After cells were incubated with the antibody at 4°C overnight, next day the samples were submitted for automated CUT&RUN which was performed by the Genomics and Bioinformatics Center at Fred Hutch on a BioMek platform.
2.  Used nuclei instead of cells: H3K4me1 and H3K4me2 were mapped using the CUT&RUN protocol as previously described using isolated nuclei (*Skene and Henikoff, 2017*).

A summary of all CUT&RUN samples with conditions and methods used can be found at Additional *Source data 1*.

## CUT&RUN data analysis

Basic analysis: Sequencing reads were aligned to hg19 using Bowtie2: bowtie2 `--end-to-end --very-sensitive --no-overlap --no-dovetail --no-unal --no-mixed --no-discordant` -q -I 10X 700x path/to/Bowtie2/indices –1 read1.fastq.gz –2 read2.fastq.gz.

> CPM normalized bigwig files were generated using bedtools genomecov.
> Peaks were called using MACS2. Peak calling was performed for each target with and without the IgG control.
> Narrow peaks with IgG control: macs2 callpeak `--name  TARGET --treatment` path/to/TARGET/hg19.bam `--control` path/to/IgG/hg19.bam `--format  BAMPE  --gsize  hs --keep-dup` all -q 0.05
> Narrow peaks without IgG control: macs2 callpeak `--name TARGET --treatment` path/to/TARGET/hg19.bam `--format BAMPE --gsize hs --keep-dup` all -q 0.05

IgG-controlled peaks that overlap with no-control peaks were retained for further analyses. For each TF/histone mark mapped in each genotype, peaks from three replicates were considered to make a final peak-set to use for downstream analysis.

Differential binding analysis: Merged peak set for each target was used for the analysis. Feature-Counts (*Liao et al., 2014*) in Subread 1.6.5 was used to count reads mapped to merged peaks in each sample. Bioconductor package edgeR 3.26.8 (*Robinson et al., 2010*) was used to detect differential peaks between conditions. Peaks with low read numbers were excluded using edgeR function filter-ByExpr with min.count=10 and min.total.count=15. The filtered count matrix was normalized by TMM method (*Robinson and Oshlack, 2010*) and subjected to significance testing using generalized linear model and quasi-likelihood method. Peaks were deemed differentially bound if absolute fold changes

were above 1.5 and FDRs were less than 0.01 for H3K27ac and Jun data, and FDR 0.05 for Myc data. Differential sites for H3K27ac, Jun, and Myc are provided as source data.

## Other data processing, analysis, and visualization

1. Correlation between RNA-Seq and the distribution of histone marks around TSSs.

A reference list of hg19 genes was downloaded from the UCSC Table Browser. Genes were oriented according to the directionality of gene transcription and specified a 2-kb window around TSSs. Genes that have an overlapping TSS within the 2-kb window and mitochondrial genes were removed, creating a list of 22,222 TSSs. The gene list was sorted in descending order of their RNA-Seq FPKM values. CUT&RUN H3K27ac and H3K27me3 signal (merged from three replicates) were mapped onto the ordered genomic sites. The coverage of histone marks was quantified using bedtools coverage and converted to FPKM values. Correlation between RNA-Seq and histone mark FPKM values was calculated using R cor.test function (method=spearman).

2. Correlation matrices were generated using deepTools (*Ramírez et al., 2016*).
3. Gene set enrichment analysis (Gene Ontology terms) was done using the Enrichr web-based tool (*Kuleshov et al., 2016*). Upregulated and downregulated genes (FDR≤0.05) in Fbw7 mutant cells were separately run through Enrichr. Enriched gene sets with Adjusted $p<0.05$ were extracted and then manually curated to remove redundant gene sets with >40% overlap. Complete output and summarized gene sets are in *Figure 1—source data 3* (Hct116) and *Figure 6—source data 2* (NSCs).
4. Motif identification. For all motif analysis, we used the MEME Suite (*Bailey et al., 2009*). We used bedtools getfasta to generate FASTA files for genomic sites of interest (*Quinlan and Hall, 2010*). For motif discovery analysis we submitted the center 100-bp sequence of peaks to MEME-ChIP. MEME-ChIP was used with default parameters in Classic mode. HOCOMOCO Human (v11 FULL) motif database was used. We used the position-weight matrix of the motif discovered by MEME-ChIP as the input for FIMO, to quantify the abundance of the motif. We used FIMO with a threshold value of p≤0.01 to capture all motif configurations and then filtered the output to select only the motifs with the highest FIMO motif scores (the higher the score, the more similar to the input motif). For differential motif analysis, we used MEME-ChIP in Differential Enrichment mode with default parameters.
5. Annotations. To assign gene regions where peaks are located, we used ChIPseeker, a R/Bioconductor package (*Yu et al., 2015*). We used the nearest gene method to assign a peak to a gene using the bedtools closest tool (*Quinlan and Hall, 2010*). Gencode Human Release 31 (GRCh37) Comprehensive gene annotation list was used to generate a list of genes with full gene coordinates which was used to annotate peaks to the nearest gene.
6. Data visualization. Plots were generated using R (https://www.r-project.org) (*R Development Core Team, 2020*). Heatmaps were generated using Deeptools (*Ramírez et al., 2016*).
7. Venn diagrams. Intersection between genomic sites was generated using Intervene Venn module (*Khan and Mathelier, 2017*).
8. Primary cancer and cell line data analysis. CIITA expression data from Fbw7 WT and mutated colon and rectal cancers were collected from the TCGA COADREAD database via UCSC Xena browser (*Goldman et al., 2020*; *Figure 5—source data 2*). CIITA expression in Fbw7 WT and mutated CRC cell lines were collected from the DepMap Portal (https://depmap.org/portal/) (*Barretina et al., 2012*). Statistical analysis was performed on GraphPad Prism. Unpaired t-test (two-tailed) was used to determine the statistical significance of CIITA differential expression of TCGA and CCLE data sets.
9. Bigwig files (three replicates merged) were viewed on Integrative Genome Viewer to show examples of CUT&RUN binding data as peaks. Schematic figures were created with BioRender.com.

## Antibodies

All antibodies are listed in the Key resources table.

## Cell culture

Hct116 cells were grown in Dulbecco's modified Eagle's medium (DMEM) with 10% fetal bovine serum (FBS) and PenStrep. For CUT&RUN and RNAseq experiments $2\times10^6$ cells were plated per 10-cm dish 2 days prior to harvesting. Cells were harvested using Accutase. Human fetal tissue derived U5

NCSs were cultured in NeuroCult NS-A basal medium (Stemcell Technologies) supplemented with N2 (made in-house 2× stock in Advanced DMEM/F-12 [Thermo Fisher Scientific]), B27 (Thermo Fisher Scientific), antibiotic-antimycotic (Thermo Fisher Scientific), Glutamax (Thermo Fisher Scientific), EGF, and bFGF (PeproTech). NSCs were cultured in Laminin-coated plates. Accutase was used to harvest cells for experiments. Genetically engineered NSCs (Fbw7$^{-/-}$) did not undergo any post-nucleofection selection and were cultured for less than 2 weeks prior to harvesting for experiments. The identity of Hct116 cells was authenticated by STR profiling and cells tested negative for Mycoplasma contamination by PCR assay.

## Chromatin fractionation

Untreated and Bortezomib treated (0.5 µM for 10 hr) cells were harvested and counted. Cells were resuspended in 50-µl CSK buffer (10 mM HEPES pH 6.8, 100 mM NaCl, 1 mM EGTA, 1 mM EDTA, 2 mM MgCl$_2$, 300 mM Sucrose, 0.1% Triton X-100, and Protease inhibitor—50 µl per million cells) (*Kim et al., 2008*). Cells were allowed to lyse for 5 min on ice and centrifuged for 5 min at 4°C at 1500*g*. The supernatant, which is the soluble fraction (S), was removed to a new tube. The pellet was resuspended in 1 ml of CSK buffer, then centrifuged for 5 min at 4°C at 1500*g*. The supernatant was thoroughly removed. Next, NP40 buffer with protease inhibitor and 250 U/ml benzonase was added to the cell pellet (same volume as CSK buffer was used to lyse cells). Cells were incubated for 30 min on ice. This was the chromatin fraction (C). Both soluble and chromatin fractions were sonicated and centrifuged to remove debris (5 min at 4°C at maximum speed). Total protein in all chromatin fractions was quantified using the Bradford assay and samples were normalized to total protein content. Equal volumes of chromatin and soluble fractions from each sample were used to immunoprecipitate Fbw7. Chromatin fractionation of Fbw7 was determined by >3 independent experiments.

## Immunoprecipitations and western blot analysis

Whole-cell extracts (WCEs) were made by lysing cells in 0.5% NP-40 buffer with protease inhibitor cocktail (made in-house). Then WCEs were sonicated and spun to remove debris. To immunoprecipitate Fbw7 from whole or fractionated cell lysates anti-Fbw7 Bethyl A301-720A antibody and Protein A beads were added and incubated for at least 2 hr at 4°C. Beads were then washed 3× with 1-ml NP40 lysis buffer. Eluted protein was electrophoresed on 8% polyacrylamide gels and transferred to PVDF which was blotted against Fbw7 using anti-Fbw7 Bethyl A301-720A (1:1000) and HRP conjugated anti-Rabbit secondary antibody (1:10,000). Membranes treated with ECL (made in-house) were visualized on a Bio-Rad ChemiDoc Imaging System.

## PCR amplification of CIITA

RNA was isolated from Hct116 and Raji cells using the QIAGEN RNeasy Mini Kit (Cat#: 74104). cDNA was prepared using iScript Reverse Transcription Supermix (Cat#: 1708841). CIITA PIII and PIV were amplified using specific primers (PIII: F – 5′GCTGGGATTCCTACACAATGC3′, R – 5′GGGTTCTGAGT AGAGCTCAATC3′ and PIV: F – 5′GGGAGCCCGGGGAACA3′, R – 5′GATGGTGTCTGTGTCGGGT T3′) at 60°C annealing temperature for 38 cycles (*Chen et al., 2015*). GAPDH was amplified as the control (25 cycles) using primers F – 5′GGTCGGAGTCAACGGATTTG3′ and R – 5′ATGAGCCCCAG CCTTCTCCAT3′. Platinum Taq DNA polymerase was used following the manufacturer's instructions.

## Reverse transcription and quantitative real-time PCR (RT-PCR and qPCR)

RNA was isolated using the QIAGEN RNeasy Mini Kit (Cat#: 74104) following the manufacturer's protocol and quantified using a NanoDrop Spectrophotometer. About 1 µg of total RNA was reverse transcribed into cDNA using the iScript Reverse Transcription Supermix for RT-qPCR (Bio-Rad) according to the manufacturer's protocol. cDNA was quantified using Platinum SYBR Green qPCR SuperMix-UDG with Rox (Invitrogen) in the ABI QuantiStudio5 Real-Time PCR System (Thermo Fisher Scientific). Three separately extracted RNA samples per genotype were used and three replicates of each sample were included in the 384-well plate. mRNA expression levels were normalized to the housekeeping gene *Actin*. The $2^{-\Delta\Delta CT}$ method was used to calculate differential mRNA expression in mutants compared to control samples. Primers used for qPCR (5′–3′): Actin F- CACCATTG GCAATGAGCGGTTC, R-AGGTCTTTGCGGATGTCCACGT; CIITA F- GCTGGGATTCCTACACAATGC,

R- GGGTTCTGAGTAGAGCTCAATC; HLA-DRA F- CGACAAGTTCACCCCACCAGT, R- CAGGAAAA GGCAATAGACAGG; HLA-DRB F- GAGCAAGATGCTGAGTGGAGTC, R- CTGTTGGCTGAAGTCC AGAGTG; HLA-DPA F- ATCCAGCGTTCCAACCACACTC, R- CGTTGAGCACTGGTGGGAAGAA; HLA-DPB F- GTGCAGACACAACTACGAGCTG, R- CCTGGGTAGAAATCCGTCACGT.

## Flow cytometry

For live-cell FACS analysis, cells were harvested, washed once in phosphate-buffered saline (PBS), and incubated with antibodies in 1× PBS for 1 hr on ice. Ghost Dye Violet 510 was used to discriminate live cells. For FACS analysis of fixed cells, cells were processed as described previously (*Diab et al., 2020*), with the following modifications. For DNA replication analysis, cells were pulsed with 10 µM 5-ethynyl-20-deoxyuridine (EdU) for 20–60 min. Following EdU labeling, cells were harvested using TrypLE Express (Thermo Fisher Scientific) and fixed in 2% paraformaldehyde (PFA) for 20 min at room temperature, washed once with PBS, resuspended in ice-cold methanol, and allowed to sit overnight at –20°C prior to further processing. Cells were then washed twice with 1 ml BD Perm/Wash Buffer (BD Biosciences), Following a Click-iT step for EdU labeling according to manufacturer's protocol (Thermo Fisher Scientific), cells were incubated with the appropriate primary antibodies in 3% BSA in PBS for 1 hr at room temperature. Cells were washed, incubated in secondary antibody for at least 1 hr, and in DAPI (1:1000, BD Biosciences, 564907) for at least 15 min, washed again, and resuspended in Stain Buffer (FBS—BD Biosciences). Samples were run on a BD FACSCelesta or FACSymphony, visualized with FACS Express software (DeNovo), and analyzed using FlowJo software v10.6.1.

## Cell proliferation assays

$1×10^5$ Hct116 cells and $0.7×10^5$ U5 NCSs per well were seeded in 6- and 12-well plates respectively. After 24 hr, cellular confluency was monitored for up to 85 hr using an IncuCyte S3 Live-Cell Imager (Sartorius). Proliferation was measured as percent confluence at each time point normalized to percent confluence at time 0. Population doubling time was calculated using GraphPad Prism 9.3.1 (Parameters: nonlinear regression fit: exponential [Malthusian] growth).

## Generation of U5-NSC homozygous Fbw7 knockouts

A previously described protocol to generate homozygous null mutations using CRISPR-Cas9 and nucleofection was followed (*Hoellerbauer et al., 2020b*; *Hoellerbauer et al., 2020a*). Briefly, the protocol is as follows:

CRISPR sgRNA were designed using Broad Institute's GPP Web Portal. The output list of sgRNAs was manually curated to choose three sgRNAs targeting *FBXW7*. Exons 3, 4, and 9 in *FBXW7* were targeted by 5′AAGAGCGGACCTCAGAACCA3′, 5′CTGAGGTCCCCAAAAGTTGT3′, and 5′ACATTAG TGGGACATACAGG3′ guides, respectively. A control sgRNA was included 5′GTAGCGAACGTGTCC GGCGT3′. sgRNAs were purchased from Synthego.

Cas9:sgRNA RNP nucleofection: sgRNAs were reconstituted by adding 10 µL of 1× TE Buffer 1.5 nmol of dried sgRNA. A working stock of 30-µM sgRNA was used henceforth. A working stock of Cas9 (10.17 pmol/µl) was made. To prepare RNP complexes, 1.87-µl sgRNA, 1.84-µl sNLS-SpCas9-sNLS (Aldevron), and 18.29-µl SG Cell Line Nucleofector Solution (Lonza) were mixed to make a 22-µl final volume. The mixture was incubated at room temperature for 15 min to allow RNP complexes to form. To nucleofect, $1.3×10^5$ cells were harvested. The cells were washed with PBS and resuspended with RNPs. (We were able to successfully nucleofect up to $8.5×10^5$ cells with the same volume of RNPs.) Cells were electroporated using the Amaxa 4D Nucleofector X unit and program EN-138. Nucleofected cells were plated in prewarmed media.

CRISPR editing efficiency analysis: Extraction of genomic DNA, PCR amplification of target site, and efficiency analysis was done as previously described (*Hoellerbauer et al., 2020b*; *Hoellerbauer et al., 2020a*). The primer pairs used to amplify CRISPR target sites in Exon 3: 5′TCATCACACACTGTT CTTCTGGA3′ and 5′TGTCTACCCTAGAACAGCTGT3′; Exon 4: 5′TGTGTACCTGTGATCTCTGGG3′ and 5′CACCTTGCTGTGCAACCATC3′; and Exon 9: 5′ACTGCTTTCATGTCGTGTTTCC3′ and 5′AGG AAGCTGACAACACTAGCA3′. We found that the pool of three sgRNA was the most successful at deleting *FBXW7*. This was confirmed by blotting for immunoprecipitated Fbw7 in each nucleofected sample (*Figure 6—figure supplement 1*).

## Data availability

All data generated and used in this manuscript are deposited in GEO: GSE184041.

Scripts available at https://github.com/hnthirima/data_visualization (*Thirimanne, 2022a*, copy archived at swh:1:rev:bf29ae485e76d451f8e7642724fe514660fc55df) and https://github.com/hnthirima/hierarchical_clustering (*Thirimanne, 2022b*, copy archived at swh:1:rev:a6d7bf35698dddc17e05a37f8f40cab0f5eb675f).

The results shown here are in part based upon data generated by the TCGA Research Network: https://www.cancer.gov/tcga.

## Acknowledgements

This research was supported by the following grants: NCI/NIH T32 CA080416 (HNT), NCI/NIH R01 CA215647 (BEC), R01 HG010492 (SH), R01NS119650 and R01 CA190957 (PJP), and the Genomics & Bioinformatics Shared Resource of the Fred Hutch/University of Washington Cancer Consortium (P30 CA015704).

The authors thank Jeff Delrow, Matthew Fitzgibbon, Alyssa Dawson, and Philip Corrin in the Genomics Shared Resources at Fred Hutchinson Cancer Research Center for support with experimental design, helpful discussions, library preparation, and sequencing. The authors thank Markus Welcker for helpful discussions and critical reading of the manuscript. The authors thank past and present Henikoff lab members including Jorja Henikoff, Christine Codomo, Michael Meers, Jay Sarthy, Terri Bryson, and Peter Skene for helpful discussions regarding data analysis, reagents, library preparation, and sequencing. The authors also thank Pia Hoellerbauer in the Paddison lab for assistance with neural stem cell culturing and knockout generation.

## Additional information

### Funding

| Funder | Grant reference number | Author |
| --- | --- | --- |
| National Cancer Institute | T32 CA080416 | H Nayanga Thirmanne |
| National Cancer Institute | R01 CA215647 | Bruce E Clurman |
| National Cancer Institute | P30 CA015704 | Bruce E Clurman |
| National Institutes of Health | R01 HG010492 | Steven Henikoff |
| National Institutes of Health | R01NS119650 | Patrick J Paddison |
| National Cancer Institute | R01 CA190957 | Patrick J Paddison |
| Fred Hutchinson Cancer Research Center | | Bruce E Clurman |

The funders had no role in study design, data collection and interpretation, or the decision to submit the work for publication.

### Author contributions

H Nayanga Thirimanne, Data curation, Formal analysis, Funding acquisition, Investigation, Methodology, Project administration, Resources, Software, Supervision, Validation, Visualization, Writing – original draft, Writing – review and editing, ORCID: 0000-0002-8016-3031; Feinan Wu, Data curation, Formal analysis, Methodology; Derek H Janssens, Investigation, Methodology, Supervision, Writing – review and editing; Jherek Swanger, Investigation, Writing – review and editing; Ahmed Diab, Formal analysis, Investigation; Heather M Feldman, Investigation; Robert A Amezquita, Formal analysis, Investigation, Methodology; Raphael Gottardo, Methodology, Supervision; Patrick J Paddison, Funding acquisition, Methodology, Resources, Supervision; Steven Henikoff, Bruce E Clurman, Data

curation, Funding acquisition, Methodology, Project administration, Resources, Supervision, Writing – original draft, Writing – review and editing

**Author ORCIDs**
H Nayanga Thirimanne ⓘ http://orcid.org/0000-0002-8016-3031
Derek H Janssens ⓘ http://orcid.org/0000-0003-1079-9525
Steven Henikoff ⓘ http://orcid.org/0000-0002-7621-8685
Bruce E Clurman ⓘ http://orcid.org/0000-0002-5835-9361

**Decision letter and Author response**
Decision letter https://doi.org/10.7554/eLife.74338.sa1
Author response https://doi.org/10.7554/eLife.74338.sa2

## Additional files

**Supplementary files**
• Transparent reporting form

• Source data 1. Summary of all CUT&RUN experiments. Experimental conditions of all CUT&RUN experiments included in the study.

**Data availability**
RNA-Seq and CUT&RUN sequencing data have been deposited in GEO under accession code GSE184041. All data generated or analyzed during the study are included in the manuscript and supporting files. Supplementary figures are included as a separate PDF. Source Data files have been provided for Figures 1, 2, 3, 5 and 6. Computational tools used in the study are mentioned in the Materials and Methods section. Code used are available at https://github.com/hnthirima/data_visualization (copy archived at swh:1:rev:bf29ae485e76d-451f8e7642724fe514660fc55df) and https://github.com/hnthirima/hierarchical_clustering (copy archived at swh:1:rev:a6d7bf35698dddc17e05a37f8f40cab0f5eb675f).

The following dataset was generated:

| Author(s) | Year | Dataset title | Dataset URL | Database and Identifier |
| --- | --- | --- | --- | --- |
| Thirimanne HN, Wu F, Janssens D, Paddison P, Henikoff S, Clurman BE | 2022 | Genome-Scale Analysis of Transcriptional Regulation by the Fbw7 Ubiquitin Ligase | https://www.ncbi.nlm.nih.gov/geo/query/acc.cgi?acc=GSE184041 | NCBI Gene Expression Omnibus, GSE184041 |

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
