## [Editor Report]

Fbw7 functions to control the abundance of more than 2 dozen transcriptional regulators, but how this affects transcription at the global level is largely unknown. The authors employ RNA-Seq, CUT&RUN on H3K27ac/H3K27me3, and a detailed analysis of the loci affected to provide a global analysis of the effect of Fbw7 mutation on transcription in HCT116 cells as well as neural stem cells. The results reveal complex, but intriguing, results suggesting that Fbw7 mutation affects primarily Jun and Myc functions in distal regulatory regions rather than target gene promoters. Although HCT116 cells employed (WT, Fbw7-/-, and Fbw7R/+) are clonal, there is significant overlap in the two mutant lines, which suggests that a substantial fraction of the effects reflect loss of Fbw7 activity. Analogous patterns related to Jun and Myc levels at distal regulatory regions are seen in the neural stem cells, where there is a pool of depleted cells rather than clonal cells derived from targeted mutagenesis. Intriguingly, gene sets related to epithelial mesenchymal transition (EMT) were enriched in the upregulated transcripts in both Fbw7-/- and R/+ mutant cells consistent with the idea that Fbw7 targets EMT regulatory proteins for degradation. Additional experiments and analyses performed during revision substantially strengthened this paper.

---

## [Decision Letter]

**Decision letter after peer review:**

Thank you for submitting your article "Global and context-specific transcriptional consequences of oncogenic Fbw7 mutations" for consideration by *eLife*. Your article has been reviewed by 3 peer reviewers, one of whom is a member of our Board of Reviewing Editors, and the evaluation has been overseen by Richard White as the Senior Editor. The reviewers have opted to remain anonymous.

Required Experiments

The reviewer's agreed that the following issues need to be addressed:

1. All three reviewers noted the issue of the cell lines employed, and it is recognized that the second large-scale experiment in the NSC line provides an alternative validation approach that goes some distance to addressing the limitations to the -/- and R/+ cell lines. Can the authors address: (1) cite data on the abundance for Myc, Jun or other substrates in these cells (point 1-reviewer 1), (2) discuss genes that are differentially regulated in R/+ versus -/- cells (Reviewer 2-point 1), (3) comment on the relationship between Myc target genes involved in protein synthesis (this might be an interesting place to add a new figure panel showing overlap of these classes or proteins), (4) address the extent to which differences in cell type may underlie the precise gene expression features for HCT116 versus NSC cells, (5) given the potential for cell line drift, the authors should add a paragraph/new section in the discussion concerning the limitations of the study and the potential for cell line drift, (6) provide greater experimental detail concerning how the study was performed, especially in regards to the heat maps and data presentation (Point 1-reviewer 3), (7) Could the authors check the abundance of FBXW7 target genes (Point 2-reviewer 3).

2. Might want to think about modifying or de-emphasizing the "just enough" model, given the comments of reviewers 2 and 3.

3. The authors show asssess HLA gene expression to validate the changes observed, due to the very low FPKMs.

*Reviewer #1 (Recommendations for the authors):*

This paper examines the effect of mutation of FBXW7 on the transcriptional program in HCT116 cells. FBXW7 is a substrate specific adaptor of an SCF ubiquitin ligase, and is known to control the abundance of several transcriptional regulators including Jun and Myc. FBXW7 is also a tumor suppressor and one mutation – (R505C) – blocks substrate blocks substrate binding through a dimerization-dependent dominant negative mechanism.

Initial RNA Seq experiments demonstrated altered expression of numerous genes in FBXW7 KO or R/+ cells, including changes in expression of MHC class II protein complex genes, among others. Overall, ~10% of all expressed genes displayed changes in expression when Fbw7 was mutated. Genome wide, H3K27ac levels (by CUT&RUN) correlated with gene expression while H3K27me3 was anti-correlated. Many of the regulated genes had motifs that would be consistent with regulation by Jun/AP-1, which fits with the specificity of FBXW7 targeting. This regulation appears to reflect, at least in part, the ability of FBXW7 to interact with candidate substrates on chromatin, as the R/R homozygous point mutant that blocks substrate association failed to associate with chromatin. There was also alterations in the occupancy of Myc and Jun at specific loci but this reflected a small proportion of potential loci, indicating that the effects are specific as opposed to global. These sites were primarily localized near distal regulatory regions. The occupancy for Jun, and to a lesser extent Myc, correlated with expression for a substantial portion of the genes.

The authors drill down a bit on the MHC class II genes. The CHIP analysis of Jun and Myc are consistent with more protein being present at these promoters.

The authors propose a couple of mechanisms that could explain the complex regulation – cooperative regulation of multiple genes via a single regulatory region, or transcription factor phosphorylation only occurring at specific gene regulatory loci, which would allow only selected pools of protein to be degraded (an interesting hypothesis). Also, the coordinate effects on Jun and Myc seem very strong, and are consistent with a common regulatory mechanism.

One potential limitation of the study is that the authors employed single clones of Fbw7-/- and R/+ mutant cells. However, they also performed a completely orthogonal analysis of neural stem cells, which are known to exhibit Fbw7-dependent Jun regulation. These cells were subjected to CRISPR to inactivate Fbw7 in a pooled population. Overall there were similar effects on transcriptional impact and some of the same GO terms were altered as with Fbw7-/- cells. There was also effects on genomic loci for Jun binding that mirrored the situation with HCT116 cells in terms of being in regulatory regions outside the loci. However, the genes affected were distinct. This potentially reflects the extremely different types of cells being used. In addition, through analysis of the R/+ mutant, the authors found a substantial (35-48%) overlap in regulated Jun sites, although the magnitude of the effects was generally smaller than with the knock-out. This could potentially reflect small amounts of activity of the R/+ heterodimer or possibly low levels of WT homodimer present in these cells. It seems likely that the sites affected in both the KO and the R/+ mutants reflect the major target genes for Fbw7-mediated control. Finally, the concordance between the behavior of Myc and Jun was striking.

*Reviewer #2 (Recommendations for the authors):*

I have no major criticisms of the manuscript data or conclusions but offer a few thoughts for consideration:

1. I am not convinced the data makes a clear case for the "just enough" model because a +/- control was not included. This isn't a major issue for the study overall but without this control it is hard to say if the differences between the R/+ and -/- are due to dosage or dominant negative effects. Is there any published (or unpublished) data the authors can cite on the actual abundance effects of the R/+ mutation versus the -/+ and -/-mutations on MYC, JUN or other substrates?

2. It is noted on p16 that some genes more impacted in R/+ compared to -/- cells but there is very little discussion of these genes or their characteristics. Given the focus on the R/+ mutation, some further discussion is warranted.

3. In Figure 3, were the MYC sites enriched for protein synthetic genes, given that this is a major target class for MYC?

4. To augment the GO term analysis, it would be helpful to list example top regulated genes for main classes, e.g., for cholesterol biosynthesis. Also there appears to be no comment on the strongest differentially affected genes, e.g., Figure 1B. A brief description would give a better sense of the complexity of the signatures. Are there any obvious cancer correlations for these genes?

5. The comparison between HCT116 and the NSC line is a strong point of the manuscript and while the general scale of gene expression changes is similar, it is notable that there is little overlap between the signatures except for MHC class II genes. The authors should comment on whether this is due to intrinsic cell type effects on gene expression and TF binding (i.e., are the genes expressed and do the TFs bind in the wild type cells) or is it actually due to differential regulation of occupied sites upon mutational perturbation of FBXW7?

6. A brief mention of cell line adaptation to the respective mutations is warranted as a caveat since it is possible the profiles may have been altered as cells adapted. Is there a differential growth defect between wild type, R/+ and -/-? If so, this should be mentioned and documented. On a related note, was there any sign of an apoptotic signature? Excess MYC and JUN might have been expected to trigger an apoptotic response.

7. There is little discussion or analysis of the somewhat surprising apparent loss of TF binding and gene repression at a large number of loci. Can the authors comment on this in a bit more detail, e.g., elaborate on the Ub modification in MYC-mediated transactivation? Similarly, it isn't clear why proximal promoter binding sites for MYC and JUN seem not to be affected whereas intronic and intergenic regions are.

8. The nomenclature (Fbw7 for protein FBXW7 for gene) is a bit confusing and should be explained at the first instance. Why not use consistent names for the gene and protein?

9. FBXW7 could be described as a super-hub of sorts since it controls many hub transcription factors including JUN and MYC. This regulatory hierarchy is in part what makes the study so interesting.

10. The authors could also mention E3 redundancy as a mechanism that might temper FBXW7 loss of function profiles.

11. Dimerization of FBWX7 in the introduction should be cited both for the authors' own work in human cells and preceding work in yeast. It is important to note that not all substrates require dimerization for recognition and this help could account for differences between the -/- and R/+ mutants.

*Reviewer #3 (Recommendations for the authors):*

1. Figure 1C: The details of the experimental design are not clearly described, which hinders understanding and evaluation of this study. The heatmap appears to show three samples for each genotype (WT, R/+, -/-). Does this mean three independent cell lines were used for each genotype (i.e. biological replicate)? Or does it mean three independent sequencing of one cell line for each genotype (i.e. technical replicates)? If the latter, the authors need to be very careful in interpreting the results. This is because although recent advances in the CRISPR-Cas9 system have facilitated genome editing in cell lines, it has become increasingly clear that even in cell lines, clonal variation is significant and cannot be ignored. In this case, the differences in the transcriptional landscape may reflect mere clonal variation rather than the state of Fbw7. The same problem is also present in Figure 3F and 6A.

2. In each clone of Fbw7(R/+) and Fbw7(-/-) HCT116 cells, determine the extent to which endogenous Myc, Jun, and SREBP are increased compared to WT cells. Also, check for accumulation of NICD, KLF5, and TGIF1, which are transcription factors that are known to be affected in the intestine by loss or mutation of Fbw7. If they could also include data from Fbw7(+/-) cells, it might reinforce their claims of the "just enough" model.

3. It is questionable whether MHC class II is functional in Fbw7(-/-) HCT116 cells, given that the FPKM value of CIITA does not reach 1. The FPKM value is also very low (below 1) for half of the genes detected as MHC class II protein complex genes by GO analysis (HLA-DRB5, HLA-DPB1, HLA-DRA, HLA-DOA, HLA-DRB1, and HLA-DPA1). The authors need to prove that the gene cluster of MHC class II enriched in Fbw7(-/-) HCT116 cells is indeed expressed at the protein level and is functional.

4. The most dramatic change in this paper is the enrichment of extracellular matrix organization in Figure 6B, which the authors barely mention in the text. The authors should examine and discuss which substrate is responsible for this change and how it is affected.

---

## [Author Response]

Required ExperimentsThe reviewer's agreed that the following issues need to be addressed:1. All three reviewers noted the issue of the cell lines employed, and it is recognized that the second large-scale experiment in the NSC line provides an alternative validation approach that goes some distance to addressing the limitations to the -/- and R/+ cell lines. Can the authors address: (1) cite data on the abundance for Myc, Jun or other substrates in these cells (point 1-reviewer 1),

We have extensively studied substrates in Fbw7^-/-^ and Fbw7^R/+^ Hct116 cells and apologize that these citations were not clear (Grim et al., *J Cell Biol*, 2008; Welcker et al., *Genes Dev*, 2013; Davis et al., *PNAS*, 2018; Welcker et al., *Sci Adv*, 2022). Myc is stabilized in Fbw7^-/-^ Hct116 cells but exhibits only small changes in steady-state abundance because it represses its own transcription. Myc is partially stabilized in Fbw7^R/+^ cells, which likely reflects the role of Fbw7 dimers and the concerted binding of two Myc degrons with dimeric Fbw7 (Welcker et al., *Sci Adv*, 2022). Others have found that Fbw7-dependent Jun turnover in Hct116 cells is regulated by Rack, BLM, and Usp28, but we find minimal impact of Fbw7 loss on Jun abundance (Davis et al., *PNAS*, 2018) and now cite this result. We studied Jun here because its binding site was enriched in active chromatin in Fbw7^-/-^ cells. In response to this question, we re-examined Jun turnover using ^35^S-methionine metabolic pulse chases, the most rigorous approach. We again found modest Jun stabilization in Fbw7^-/-^ cells, but this has proven challenging to quantitate: the turnover is bi-phasic and the signal-to-noise quickly deteriorates despite using copious amounts of radiolabel. Jun also turns over in Fbw7^-/-^ cells through other pathways. We suspect that Jun turnover by Fbw7 is highly context-specific, which agrees with our conclusions in this paper. We do not have enough confidence in these data to include them in the paper and will not pursue this approach further.

Reviewer 3 asks about other substrates implicated in intestinal neoplasia (Notch, SREBP, TGIF, KLF5). These (and other) substrates certainly contribute to Fbw7-associated tumorigenesis, but we have not studied them here because they were not implicated in our results. Notch is minimally elevated in Fbw7^-/-^ Hct116 cells (Davis et al., *PNAS*, 2018) and cytoplasmically regulated in Fbw7β^-/-^ Hct116 cells (Sancho et al., *PLoS Biol*, 2013). TGIF abundance is increased in both Hct116 Fbw7^-/-^ and Fbw7^R/+^ cells (Davis et al., *PNAS*, 2018). We agree with the Reviewer that TGIF is likely very important in Fbw7-associated colorectal cancer. SREBP is stabilized in Fbw7^-/-^ Hct116 cells and is dimer-dependent (Welcker et al., *Genes Dev*, 2013). We have not studied KLF5, but others found it stabilized in Fbw7^-/-^ Hct116 cells (Liu et al., *JBC*, 2010) and in Fbw7^R/+^ mice (Davis et al., *J Path*, 2011). We will study the role of these substrates in the phenotypes we describe but this is beyond the paper’s scope.

We addressed this point by clarifying the requested citations (lines 231-242).

2) Discuss genes that are differentially regulated in R/+ versus -/- cells (Reviewer 2-point 1),

In response to this (and related) comments, we re-evaluated differential gene expression in Fbw7 mutant Hct116 cells (and NSCs, see below). These analyses have proven illuminating and provocative, and we thank the reviewers and editor for these suggestions. We revised the analyses to: (1) study more stringently selected differential genes, and (2) identify and compare enriched gene sets in these differential genes more comprehensively. The heat map in Figure 1C depicts gene clusters that are either uniquely differential in Fbw7^-/-^ or Fbw7^R/+^ cells, or common to both mutants. While we initially used Enrichr to analyze these unsupervised hierarchically clustered gene sets, we have now manually retrieved the statistically significant differential genes from the source data to run on Enrichr. We also separately analyzed the up- or downregulated genes in each mutant, examined classifications in addition to Gene Ontology (e.g., MSigDB, JASPAR), and included all gene sets with adjusted p-value < 0.05. Finally, because these gene set classifications are broad and overlapping, we eliminated those with >40% overlapping genes. These new analyses are summarized in new Figures 1D and 6B, and the full list of enriched sets and their specific genes are found in new Figure 1 – source data 3 and Figure 6 – source data 3.

We now appreciate that most of the enriched biological processes are common to both Fbw7^-/-^ and Fbw7^R/+^ cells. There is also surprising conservation with NSCs (see below). Furthermore, many of the same specific genes are assigned to shared gene sets in Fbw7^-/-^ and Fbw7^R/+^ cells. In contrast, the MHC Class II gene set is specific to the Fbw7^-/-^ cells. The most enriched gene sets in the upregulated genes include the p53 pathway (see Reviewer 2-point 6) and processes associated with epithelial-mesenchymal transition (EMT). While we initially reported the latter as extracellular matrix (ECM)-associated genes (based on GO), we now interpret it as reflecting EMT, which may also include related gene sets (e.g., ECM, cell mobility). Fbw7 targets several EMT regulators (e.g., Zeb 1/2, Snail, LSD1) and given the conservation across Hct116 and NSC systems, EMT may be a more important aspect of Fbw7 function and its loss in cancer than we had previously realized. There are fewer enriched gene sets in the downregulated differential transcripts in Fbw7 mutant cells, which include TNF/NF-κB (perhaps associated with regulation of p100 NF-κB2 by Fbw7), signaling pathways (e.g., estrogen), and inflammation, which may be related to the IFN/dsRNA inhibition reported by Gstalder et al., (*Can Disc*, 2020).

As suggested by Reviewer 2 and Reviewer 3, we also looked for Fbw7 TF substrates that might underlie these pathways. The motif for KLF TFs was highly enriched in the upregulated genes, but not in downregulated genes (new Figure 1D and 6B). Several KLF proteins are Fbw7 substrates (KLF2/5/10/13); these proteins share a consensus binding site and function as repressors or activators. These data may further implicate the KLF family as an important mediator of Fbw7 function, and the motif’s absence in the downregulated genes may implicate KLF transcriptional activators. Despite the Fbw7^R/+^-specific differential transcripts (clusters 5 and 6, Figure 1C), we did not find gene sets enriched in only Fbw7^R/+^ cells. Keeping in mind the many caveats inherent in these types of analyses, the enriched gene sets shared between Fbw7^-/-^ and Fbw7^R/+^ cells may reflect common consequences of the two types of Fbw7 mutations. We considered the 50 most highly differential genes in Fbw7^R/+^ cells as suggested but did not find any obvious insights.

We responded to this point by: (1) performing new and extended Enrichr analyses (Figures 1C and 6B, Figure 1 – source data 3, Figure 6 – source data 3), and (2) multiple text revisions (lines 135-162, 391-405, 477-500).

3) Comment on the relationship between Myc target genes involved in protein synthesis (this might be an interesting place to add a new figure panel showing overlap of these classes or proteins),

We specifically examined expression and Myc occupancy of Myc target genes with roles in protein synthesis (Table 1, Ruggero, *Cancer Res*, 2009), but they were unaffected by Fbw7 status. Given the small proportion of Myc binding sites impacted by Fbw7 loss (and even fewer that could be linked with genes), this may not be surprising. Of note, we previously described the metabolic consequences of Fbw7 loss in Hct116 cells (Davis et al., *PNAS*, 2018). In contrast to Myc-associated Warburg metabolism, Fbw7^-/-^ and Fbw7^R/+^ cells exhibited increased oxidative phosphorylation without activation of Myc metabolic target genes, consistent with the current conclusion that Myc function is not globally augmented by Fbw7 loss (in these cells).

We responded to this point with new text in the Discussion (lines 457-461).

4) Address the extent to which differences in cell type may underlie the precise gene expression features for HCT116 versus NSC cells,

We addressed this point in two ways. First, as described above, we compared the enriched gene sets in Fbw7 mutant NSCs and Hct116 cells. Despite the differences between the two cell systems (e.g., stable versus acute Fbw7 deletion, cell type, transformed versus primary, clonal versus non-clonal), they shared many enriched pathways, suggesting substantial conservation of Fbw7-dependent processes (new Figure 6B, Figure 6 – source data 3). This concept is perhaps the most significant change in our revision and may open new and important directions. Second, we used hierarchical clustering to compare the differentially expressed genes in NSCs and Hct116 cells (new Figure 6 —figure supplement 6A, 6B). This analysis showed that: (1) some differential genes in one cell type are not expressed/captured in the other cell type, and (2) cell-type specific differential genes were the most differential genes within that cell type (e.g., differential genes specific to Hct116 cells had the highest log_2_FC compared to differential genes expressed in both Hct116 and NSCs (Wilcoxon test: p-value = 5.33e-16)).

We responded to this point by: (1) performing new gene set enrichment analyses and comparisons with the Hct116 cells (Figure 1C and Figure 6B), (2) comparing the differential genes in Hct116 and NSCs (Figure 6 —figure supplement 6), and (3) with new text (lines 135-162, 391-405, 477-500).

5) Given the potential for cell line drift, the authors should add a paragraph/new section in the discussion concerning the limitations of the study and the potential for cell line drift,

This issue was important to all three Reviewers. We share this concern and had tried to mitigate it in several ways. As requested, we added the following paragraph to the revised Discussion:

“Finally, it is important to note the potential impact of clonal variation on our studies that involved Hct116 cell clones. Several factors increase our confidence that our findings we describe are directly attributable to Fbw7 status, rather than clonal evolution. First, we found that a large proportion of differential loci and transcripts were shared between the two Fbw7 mutant Hct116 cell lines (Figure 1C and 4A), and many of these exhibited a gradient of deregulation that tracked with the severity of the mutations (Fbw7^-/-^ > Fbw7^R/+^). Because these cells were independently derived, it is unlikely that these shared features arose stochastically through clonal evolution in each line. Next, we acutely deleted Fbw7 in the orthogonal and non-clonal NSCs to mitigate against this possibility, and the shared consequences of Fbw7 loss (proportion of differential sites and transcripts, intergenic location of differential sites, shared biological processes) in these two disparate cell systems support the conclusion that these observations reflect shared and direct consequences of Fbw7 loss. Finally, we studied biological replicates of non-clonal NSCs in which Fbw7 was independently inactivated by high efficiency nucleofection-based CRISPR/Cas9 editing, and the consistent results across these replicates indicates that the differential loci are unlikely to result from “drift” phenomena. In future studies, we plan to adapt the strategy of using high-efficiency gene targeting to acutely delete Fbw7 in primary cells, to study other tissues and stem cells. These studies will also allow us to again assess any contributions of clonality”.

We responded to this point with new text in the Discussion (lines 536-553).

6) Provide greater experimental detail concerning how the study was performed, especially in regards to the heat maps and data presentation (Point 1-reviewer 3),

For Hct116 cells, the RNA-Seq and CUT&RUN experiments included three independent replicates of one clone for each genotype (Figure 1C and 3F). Each sample was harvested from a separately cultured plate and independently processed throughout an experiment. The NSCs are primary cells. Fbw7-deleted NSC replicates were generated with two independent high efficiency CRISPR/Cas9 nucleofections that did not involve selection or cloning. NSCs were cultured for less than 2 weeks prior to use for the experiments.

We responded to this point with new text Methods (lines 561-565, 592-595) and legends in Figures 1, 3 and 6.

7) Could the authors check the abundance of FBXW7 target genes (Point 2-reviewer 3).

We clarified with the Reviewing Editor that this question refers to the substrates that we have discussed in point 1.

2. Might want to think about modifying or de-emphasizing the "just enough" model, given the comments of reviewers 2 and 3.

In hindsight, we inadvertently emphasized the “just enough” model in this paper. We did not perform this study to address this model. Rather, the differential transcripts and loci in both mutant genotypes that were intermediately impacted in the missense cells reminded us of the “just-enough model”. That is, they seemed to be molecular examples of the reduced severity of the missense mutation, which is the crux of “just enough”. We qualified the revised text and deleted the statement supporting “just enough” from the introduction (original lines 106-108).

We responded to this point with text edits to de-emphasize this point (deleted intro text, new text lines 464-468).

3. The authors show asssess HLA gene expression to validate the changes observed, due to the very low FPKMs.

We addressed this in several ways. First, we performed the requested qPCR in both cell types, which closely matches and validates the HLA class II gene expression seen in the RNA-Seq data (new Figure 5C and new Figure 6F). We also used flow cytometry to detect Class II HLA proteins on the surface of Hct116 cells, which revealed increased, but heterogeneous, expression (Figure 5 —figure supplement 2, Figure 6 —figure supplement 5). This heterogeneity could reflect either assay sensitivity or variable expression, and it is also seen in NSCs. We do not yet understand the reason for the heterogeneous expression. We recently performed a single cell RNA-Seq experiment in NSCs, which suggested that CIITA expression is heterogeneous. We will pursue these studies, but they are outside the scope of our current paper. Finally, immunoblotting revealed increased HLA Class II protein expression after Fbw7 deletion in both cases (Figure 5 —figure supplement 2, Figure 6 —figure supplement 5).

We responded to this point with: (1) new quantitative RT-PCR data (Figure 5C and 6F) (2) new flow cytometry and immunoblotting data (Figure 5 —figure supplement 2, Figure 6 —figure supplement 5), and (3) new text (lines 343-353, 417-418).

Reviewer #1 (Recommendations for the authors):This paper examines the effect of mutation of FBXW7 on the transcriptional program in HCT116 cells. FBXW7 is a substrate specific adaptor of an SCF ubiquitin ligase, and is known to control the abundance of several transcriptional regulators including Jun and Myc. FBXW7 is also a tumor suppressor and one mutation – (R505C) – blocks substrate blocks substrate binding through a dimerization-dependent dominant negative mechanism.Initial RNA Seq experiments demonstrated altered expression of numerous genes in FBXW7 KO or R/+ cells, including changes in expression of MHC class II protein complex genes, among others. Overall, ~10% of all expressed genes displayed changes in expression when Fbw7 was mutated. Genome wide, H3K27ac levels (by CUT&RUN) correlated with gene expression while H3K27me3 was anti-correlated. Many of the regulated genes had motifs that would be consistent with regulation by Jun/AP-1, which fits with the specificity of FBXW7 targeting. This regulation appears to reflect, at least in part, the ability of FBXW7 to interact with candidate substrates on chromatin, as the R/R homozygous point mutant that blocks substrate association failed to associate with chromatin. There was also alterations in the occupancy of Myc and Jun at specific loci but this reflected a small proportion of potential loci, indicating that the effects are specific as opposed to global. These sites were primarily localized near distal regulatory regions. The occupancy for Jun, and to a lesser extent Myc, correlated with expression for a substantial portion of the genes.The authors drill down a bit on the MHC class II genes. The CHIP analysis of Jun and Myc are consistent with more protein being present at these promoters.The authors propose a couple of mechanisms that could explain the complex regulation – cooperative regulation of multiple genes via a single regulatory region, or transcription factor phosphorylation only occurring at specific gene regulatory loci, which would allow only selected pools of protein to be degraded (an interesting hypothesis). Also, the coordinate effects on Jun and Myc seem very strong, and are consistent with a common regulatory mechanism.One potential limitation of the study is that the authors employed single clones of Fbw7-/- and R/+ mutant cells. However, they also performed a completely orthogonal analysis of neural stem cells, which are known to exhibit Fbw7-dependent Jun regulation. These cells were subjected to CRISPR to inactivate Fbw7 in a pooled population. Overall there were similar effects on transcriptional impact and some of the same GO terms were altered as with Fbw7-/- cells. There was also effects on genomic loci for Jun binding that mirrored the situation with HCT116 cells in terms of being in regulatory regions outside the loci. However, the genes affected were distinct. This potentially reflects the extremely different types of cells being used. In addition, through analysis of the R/+ mutant, the authors found a substantial (35-48%) overlap in regulated Jun sites, although the magnitude of the effects was generally smaller than with the knock-out. This could potentially reflect small amounts of activity of the R/+ heterodimer or possibly low levels of WT homodimer present in these cells. It seems likely that the sites affected in both the KO and the R/+ mutants reflect the major target genes for Fbw7-mediated control. Finally, the concordance between the behavior of Myc and Jun was striking.

We are also interested in the function of Fbw7β and even made Fbw7β^-/-^ mice, which have no phenotype (yet). We searched the genes in the ER term for shared TF substrates but didn’t find any. We also expect that a TF substrate would be targeted in the nucleus by Fbw7α.

Reviewer #2 (Recommendations for the authors):4. To augment the GO term analysis, it would be helpful to list example top regulated genes for main classes, e.g., for cholesterol biosynthesis. Also there appears to be no comment on the strongest differentially affected genes, e.g., Figure 1B. A brief description would give a better sense of the complexity of the signatures. Are there any obvious cancer correlations for these genes?

All of the specific genes in the gene sets are shown in Figure 1 – source data 3 and Figure 6 – source data 3. Looking at the very top differential genes, however, did not prove very informative. There is some overlap in the top 50 up (11/50) and down (8/50) deregulated transcripts in Fbw7^-/-^ and Fbw7^R/+^ cells, but no obvious shared function or regulation. As discussed above, some enriched pathways are likely to be very important for cancer, such as EMT. The sterol synthesis signature is composed of genes that are modestly decreased in Fbw7^-/-^ Hct116 cells (we had initially combined the up- and downregulated transcripts).

We responded to this point with new analyses and text (lines 468-476)

6. A brief mention of cell line adaptation to the respective mutations is warranted as a caveat since it is possible the profiles may have been altered as cells adapted. Is there a differential growth defect between wild type, R/+ and -/-? If so, this should be mentioned and documented. On a related note, was there any sign of an apoptotic signature? Excess MYC and JUN might have been expected to trigger an apoptotic response.

We initially made these cells to study the cell cycle and should have discussed this point. We don’t see apoptotic responses in either the Hct116 or NSCs, probably because Myc and Jun are not grossly overexpressed. p53-p21 induction is the major homeostatic response to Fbw7 loss. It is strongest in primary fibroblasts (Minella et al., *Oncogene* 2008; Minella et al., *Current Biology*, 2002) and occurs in tumors (Li et al., *Oncogene*, 2015). Cyclin E-CDK2 hyperactivity may be the primary signal to p53 through replication stress. Excess cyclin E activity also inhibits S-phase progression in Fbw7 null cells through its impact on replication dynamics.

New Figure 1 —figure supplement 2 shows that Fbw7 mutations in Hct116 cells increase doubling time by 20-25% and modestly increase S-phase fraction (Fbw7^-/-^ > Fbw7^R/+^). We also see ~two-fold increase in p21 mRNA in both Fbw7 mutant Hct116 cell lines in the RNA Seq data. Figure 6 —figure supplement 2 shows that Fbw7 deletion in NSCs similarly increases doubling time but with a small decrease in S-phase fraction. We do see enriched gene sets associated with p53 in the Fbw7 mutant cells but not any reflective of significant cell cycle disruption, DNA damage, or apoptosis. It seems unlikely that the cell cycle changes inherent to Fbw7 loss account for any of the transcriptional signatures, except for perhaps the p53 pathway (although many of the nominally “p53 pathway” genes are only loosely associated with p53, *per se)*. However, we cannot exclude this possibility, which we now discuss.

We responded to this point with new analyses of proliferation in Hct116 cells and NSCs (Figure 1 —figure supplement 2, Figure 6 —figure supplement 2 and new text lines 146-147, 387-388, 496-500).

7. There is little discussion or analysis of the somewhat surprising apparent loss of TF binding and gene repression at a large number of loci. Can the authors comment on this in a bit more detail, e.g., elaborate on the Ub modification in MYC-mediated transactivation? Similarly, it isn't clear why proximal promoter binding sites for MYC and JUN seem not to be affected whereas intronic and intergenic regions are.

These are important questions that we just don’t know the answers to yet. In the case of decreased TF occupancy in Fbw7^-/-^ cells, our best guess is that binding partners or other proteins that regulate TF binding (e.g., the CDK8 Mediator subunit) are also affected by Fbw7 mutations, but we don’t have any experimental evidence for this idea. Nor do we understand why the promoter proximal sites are relatively insensitive to Fbw7 status. We also suspect that the answer for Myc may lie somewhere in the relationship between Myc ubiquitylation and transcriptional processivity, but these ideas are too nascent to discuss at present.

We responded to this point with new text in the Discussion (lines 452-457).

8. The nomenclature (Fbw7 for protein FBXW7 for gene) is a bit confusing and should be explained at the first instance. Why not use consistent names for the gene and protein?

The gene name is FBXW7, and the protein is correctly called either Fbxw7 or Fbw7. Because we used “Fbw7” in other papers, we kept this practice for consistency, and explain the nomenclature in the revision.

10. The authors could also mention E3 redundancy as a mechanism that might temper FBXW7 loss of function profiles.

Agreed- E3 redundancy is an important factor that we now mentioned in the revision, as suggested (lines 67-69).

11. Dimerization of FBWX7 in the introduction should be cited both for the authors' own work in human cells and preceding work in yeast. It is important to note that not all substrates require dimerization for recognition and this help could account for differences between the -/- and R/+ mutants.

Thank you. As noted, Fbw7 dimerization is a major focus of our work, including a recent paper describing two Myc degrons that bind cooperatively to dimeric Fbw7 (Welcker et al., Sci Advances, 2022). As noted, dimerization is important for other CRLs in mammals (e.g., β-TrCP, SPOP, Keap1) and for Fbw7 orthologues (e.g., pop1/2). We expanded our discussion of dimerization, substrate specificity, and R/+ mutations, including with respect to Fbw7 orthologues.

Reviewer #3 (Recommendations for the authors):4. The most dramatic change in this paper is the enrichment of extracellular matrix organization in Figure 6B, which the authors barely mention in the text. The authors should examine and discuss which substrate is responsible for this change and how it is affected.

Thank you for this comment; it prompted us to re-examine the gene set enrichment analyses. We had not appreciated the possible significance of the ECM gene set enrichment and its conservation across Hct116 and NSCs. As discussed above, we now believe this gene set to be related to EMT, which includes the deposition of ECM proteins and affects other properties such as migration and adherence. We have not found evidence for which TF(s) may be responsible for the expression of the EMT genes. There are several Fbw7 substrates that regulate EMT, including Zeb 1/2. Snail, and LSD1. We have not studied these substrates ourselves but plan to do so. Another possibility involves the PRC2 complex, a major EMT regulator, since EZH2PRC2 is regulated by cyclin E-CDK2. We can assure Reviewer 3 that this has risen high on our priority list and may turn out to be the most significant finding of our study.

We responded to this point with new text in the Discussion (lines 147-155, 391-399, 477-500).